# The Viability of a Grid of Autonomous Ground-Tethered UAV Platforms in Agricultural Pest Bird Control

Joshua Trethowan, Zihao Wang and K. C. Wong *

School of Aerospace, Mechanical and Mechatronic Engineering, The University of Sydney, Sydney, NSW 2006, Australia
* Correspondence: kc.wong@sydney.edu.au; Tel.: +61-2-9351-2347

**Abstract:** Pest birds are a salient problem in agriculture all around the world due to the damage they can cause to commercial or high-value crops. Recent advancements in Unmanned Aerial Vehicles (UAVs) have motivated the use of drones in pest bird deterrence, with promising success already being demonstrated over traditional bird control techniques. This paper presents a novel bird deterrence solution in the form of tethered UAVs, which are attached and arranged in a grid-like fashion across a vineyard property. This strategy aims to bypass the power and endurance limitations of untethered drones while still utilising their dynamism and scaring potential. A simulation model has been designed and developed to assess the feasibility of different UAV arrangements, configurations, and strategies against expected behavioural responses of incoming bird flocks, despite operational and spatial constraints imposed by a tether. Attempts at quantifying bird persistence and relative effort following UAV-induced deterrence are also introduced through a novel bird energy expenditure model. This aims to serve as a proxy for selecting control techniques that reduce future foraging missions. The simulation model successfully isolated candidate configurations, which were able to deter both single and multiple incoming bird flocks using a centralised multi-UAV control strategy. Overall, this study indicates that a grid of autonomous ground-tethered UAV platforms is viable as a bird deterrence solution in agriculture, a novel solution not seen nor dealt with elsewhere to the authors' knowledge.

**Keywords:** unmanned aerial vehicle; UAV; tethered UAV; pest bird control; drone; intelligent system; path planning

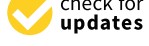



## 1. Introduction

Damage caused by birds is a long-standing and costly problem in agricultural sectors across the world, especially in high-value crops such as blueberries, cherries and wine grapes [1–4]. The United States—considered one of the top five agriculture-producing countries globally [4]—experienced more than USD 800 million in losses in 2004 solely due to European Starlings, with total losses from all bird species estimated in excess of USD 4.7 billion [5]. The heavy damage is attributed to the fact that most of the existing bird damage control methods have not proven to be effective [2].

With improvements in UAV platforms and autonomous technologies, several studies have explored the use of UAVs, in particular multirotor type UAVs, as an effective form of bird deterrence through pursuit and shepherding manoeuvres [6–9]. Wang et al. in [7] utilised a novel predator multirotor UAV system and demonstrated successful bird deterrence up to a 50 m radius centred on the UAV itself. Similarly, a study in [10] used a quadcopter system to deter birds from a vineyard in the United States and concluded that random patterns and distress calls could significantly reduce bird foraging activity in the region, with up to a 50% reduction during flight cycles compared to controlled no-flight cycles over a 14-day period. Anecdotally, many farmers have also started using commercial off-the-shelf UAVs for bird deterrence with some success. However, these

UAVs still require human operators throughout their mission to be successful. Evidently, for a bird deterrence system to be cost-effective, it must aim to be autonomous and mitigate any additional manual labour. If the user is required to dedicate a large portion of their time using the bird control system, then they lose productive time that could be used elsewhere on the property.

A major limitation of current multirotor UAV technologies is their limited flight time and endurance. Most typical small multirotor UAVs have an endurance of approximately 10 to 30 min before they need to recharge [11]. This directly conflicts with the established aim of mitigating additional manpower since an operator would be required at frequent intervals throughout the day to recharge the UAV or replace its battery.

A potential solution to this problem is using a power-over-tether multirotor system. Recent advancements in tether technology have enabled capabilities such as the ground stations in [12–15], with the one in [12] autonomously varying a 12 m long tether, using a winch system to ensure that the tether is kept taut without excessively pulling on the multirotor and impacting its dynamics. Compared to a free-flying multirotor with limited endurance, such a system would have indefinite endurance through power delivered via their physical tethers from a ground station. The tether would also improve safety by physically restricting the flying vehicles' movements, preventing collision accidents, and removing fly away concerns, which would make the system more likely to be approved by regulators.

The objective of this paper is to explore the feasibility of using multiple multirotor UAVs that are tethered to the ground in a grid-like manner to deter bird flocks from a given property. The project builds on the prior study in [7,8,16] to develop a simulation model and investigate the suitable UAV arrangements, configurations, and strategies against expected behavioural responses of incoming bird flocks.

## 2. Simulation Model and Methodology

This work extends and adapts the trajectory planning algorithm based on probability maps as well as the bird behaviour model based on field trials used by Wang [17]. The focus of this paper is placed on the multi-UAV arrangement and coordination due to dynamic and spatial constraints imposed by finite tether lengths. Modelling and predicting bird behaviour is notoriously difficult; thus, simplifications and assumptions must be made on the three key components which define this problem: the property or 'world' model, the tethered UAV model (hereon 'agent'), and the bird flock model (hereon 'target' and 'target flock').

### 2.1. Property World Set-up

Observations of typical vineyard plot sizes in Australia have been found to be approximately 100 m by 100 m in size. Hence, to simplify this problem, it will be assumed throughout the simulation that the property 'world' is controlled as a square of these dimensions.

The property itself is modelled as a probability map or occupancy grid, similar to those described in [16,18]. This method models the search area as a grid of cells wherein each cell is given a single value representing the probability of a given target occupying it based on external sensor information. In the problem of UAVs deterring target flocks, this is an appropriate scheme to adopt as the precise location of the target is unknown representative of search-and-capture or pursuit–evasion missions. Hence, UAV agents chase targets by pursuing regions of high probability despite lacking knowledge of the true state of the target.

Though the real-life problem involving UAVs and birds is inherently 3D, from field trials in [7], birds that pass well above the ground are of no concern to the crop being protected, while UAV altitude ranges from ground level to approximately 15 m have shown comparable success in initiating flight responses or evasive manoeuvres in targets within the same height range [7]. Hence, the occupancy grid map is modelled in 2D to reduce computational complexity. Despite this, aspects of 3D dynamics are still implemented elsewhere within the agent and target models.

Each cell within the simulation property lies within the horizontal dimensions $[x_{min}, x_{max}]$ in $x$ and $[y_{min}, y_{max}]$ in $y$ to form the spatial domain $M$, defined as:

$$M = \left\{ \bar{c} \,\middle|\, \begin{array}{l} \bar{c}_x \in [x_{min}, x_{max}] \\ \bar{c}_y \in [y_{min}, y_{max}] \end{array} \right\}, \tag{1}$$

where $\bar{c}$ is a vector representing the centre coordinates of each individual grid cell.

From the square property assumption earlier, it is intuitive to use square-shaped cells of edge length $L$ to populate a discrete $N \times N$ cell set grid map. The modified spatial domain and centre coordinates of each cell is therefore defined as:

$$\widetilde{M} = \left\{ \bar{c} \,\middle|\, \begin{array}{l} \bar{c}_x = x_{min} + L(i - 0.5) \\ \bar{c}_y = y_{min} + L(i - 0.5) \end{array} , i = 1, 2, \ldots, N \right\}. \tag{2}$$

For this project, grid cells with an edge length of $L = 5$ m were chosen to populate the 100 m by 100 m property (forming a 20 by 20 cell set), as shown in Figure 1, since a higher fidelity than this in localization is not expected with current technologies. The probability score $k$ of a cell ranges between 0 and 1, where $k = 1$ represents a 100% likelihood of a target occupying that cell while $k = 0$ is the opposite. Hence, observing Figure 1c, regions of higher probability correlate to dark grey cells, while the converse is true with light grey to white cells.

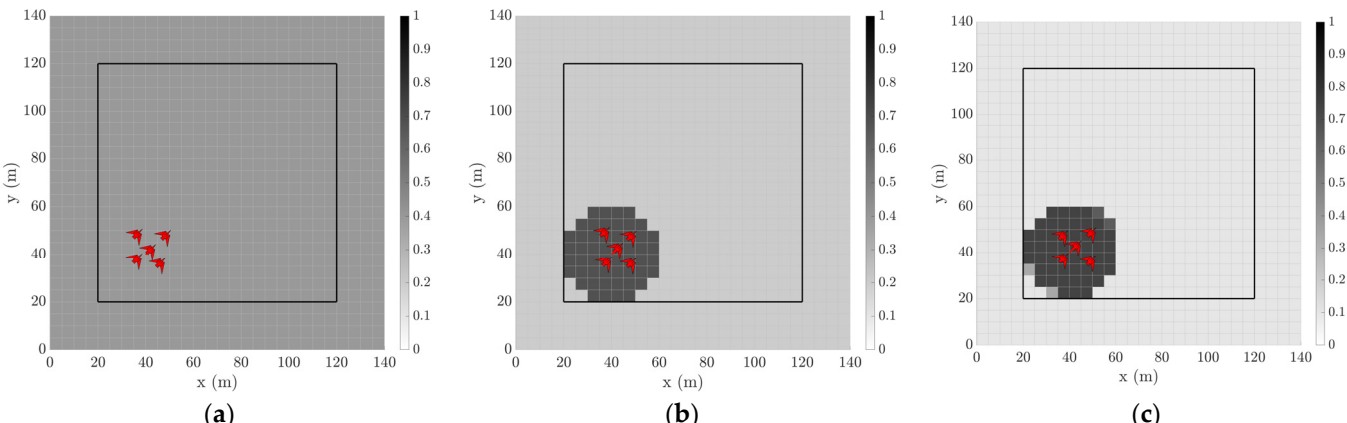

**Figure 1.** Target probability map over incremental time steps (darker cells represent higher probability of target occupancy based off sensor estimates while the red markers represent the true target flock position). Note flocks are not drawn to scale. (**a**) Probability grid at $t = 0$ s; (**b**) Probability grid at $t = 1$ s; (**c**) Probability grid at $t = 2$ s.

Importantly, this probability score is updated at every time step so the system can have continued confidence regarding the location of the target. However, if there is no sensor information, the probability score of that cell will begin approaching a non-zero nominal value (in other words, away from the certainties of $k = 0$ and $k = 1$). The nominal probability value $k_{nominal}$ is approached as:

$$k(t + 1, \bar{c}) = \tau_{probability} k(t, \bar{c}) + \left( 1 - \tau_{probability} \right) k_{nominal}, \tag{3}$$

where $\tau_{probability} \in [0, 1]$ is a time constant determining the rate that $k$ approaches $k_{nominal}$. The probability nominalisation model and the parameters are chosen to reflect the limited bird behaviour observed by the authors. Further data are needed to verify the accuracy of the model.

Each agent is equipped with a sensor model that detects targets within a circular sector of radius $r_{sensor}$ and angle $\theta_{sensor}$, as illustrated in Figure 2. All of the cells within the sector

are designated a probability score $k_{high}$ if a target is detected by the sensor. Conversely, all of the cells are designated $k_{low}$ upon zero detection. In addition, virtual ground cameras are positioned at the corners of the property to provide sensor information across the whole protected area (designated by the black box in Figure 1) that each agent has access to at every time step. Bird detection and localisation using an arrangement of ground cameras in tandem with mobile ones mounted on UAVs can be found in [19]. A camera network like this improves localisation accuracy in detecting target flocks.

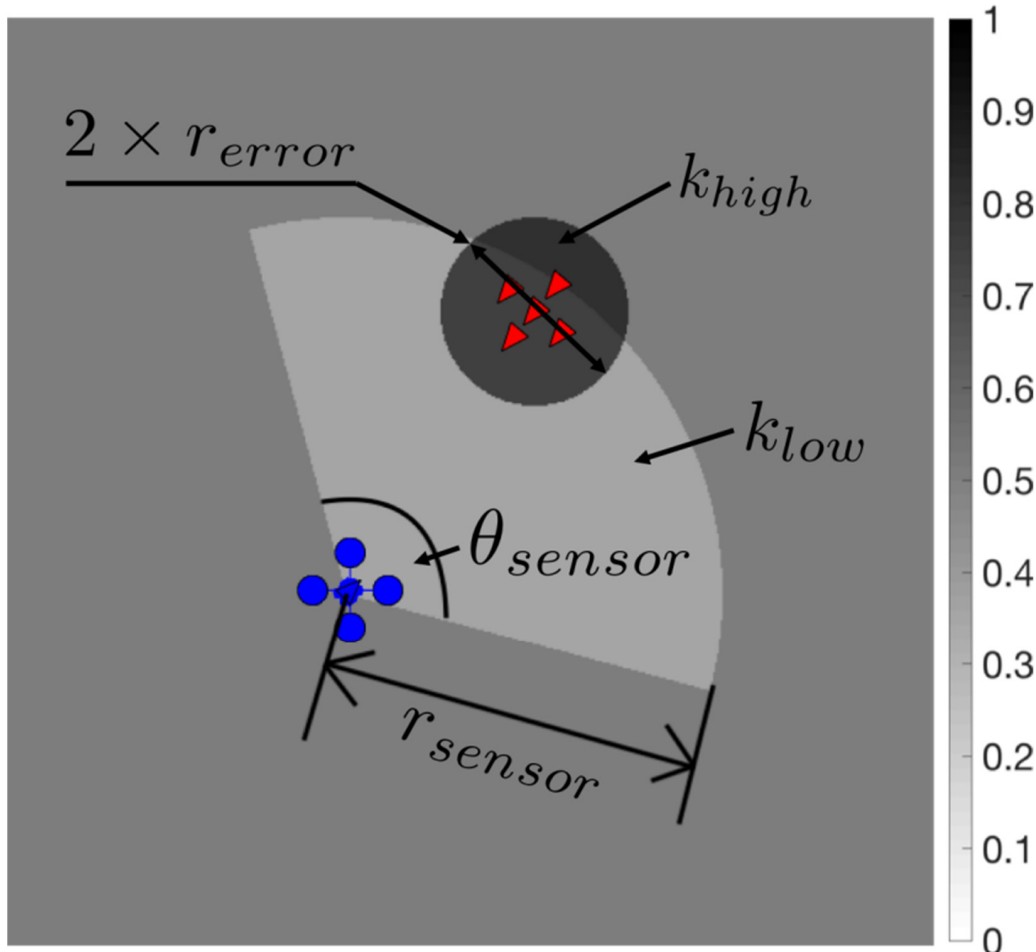

**Figure 2.** Illustration of the sensor model. The greyscale bar indicates the probability of a target occupying the cells, darker indicates higher probability.

At the initialisation of the simulation, all of the grid cells are initialised with the same nominal value before any sensor data are processed, as can be seen in Figure 1a, where each cell has a probability score of $k = 0.4$. In the next time step, shown in Figure 1b, the global probability values are updated, such that the cells near the true target position are now darker ($k = 0.7$), while the regions further away become lighter ($k = 0.2$). Finally, by the next time step, shown in Figure 1c, the colour contrast becomes more salient, indicating increased confidence in the system as to where the target flock both is and is not.

### 2.2. Tethered UAV Model

#### 2.2.1. UAV Dynamics

All of the agents in this problem are assumed to be tethered multirotor UAVs. The agents are modelled using simple second-order dynamics with the following state vector $X_{UAV}$:

$$X_{UAV} = (x, y, z, v_x, v_y, v_z, \psi), \tag{4}$$

where $(x, y, z)$ represent the position; $(v_x, v_y, v_z)$ represent the velocity; and $\psi$ represents the heading. The vertical dynamics (z-axis) have been decoupled from the horizontal dynamics (xy-plane) due to the aforementioned assumptions, where positional accuracy and modelling in height are of limited concern to the results. This similarly assumes the pitch and roll to be zero and are thus not included in the state vector as a further simplification. The horizontal dynamics are governed by the relative error between the current position of the agent and the desired waypoint and are constrained by maximum speed ($v_{max}$), acceleration ($a_{max}$), yaw rate ($\psi_{max}$), and tether length ($L_T$).

In this simulation, the effective tether length is defined as the horizontal projection along the ground, as demonstrated in Figure 3, as opposed to the true length in a 3D space, since the horizontal distance has a significantly greater impact on UAV dynamics and stability. Hence, this tether length defines a maximum radius in which an agent can move, mapped to a circle in 2D (Figure 3a) and a cylinder in 3D (Figure 3b) and will thus be hereon defined as the tether radius ($R_T$) for this simulation.

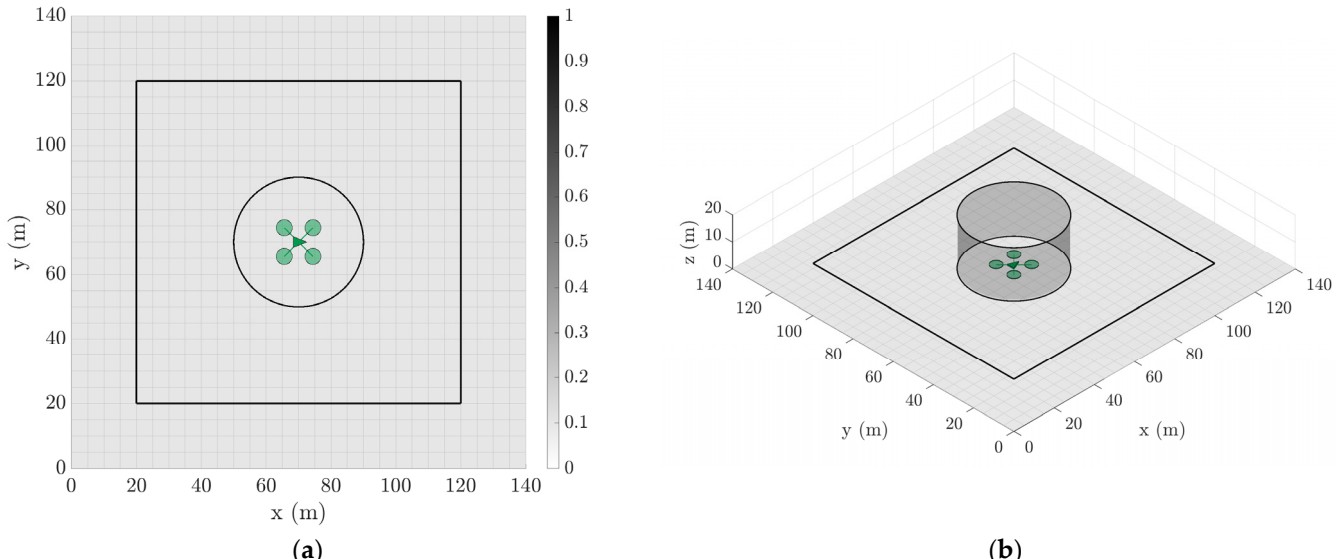

**Figure 3.** Agent UAV radius of operation sample case $R_T = 20$ m. The maximum range maps to a circle in 2D (**a**) and a cylinder in 3D (**b**). Note the UAV is not drawn to scale.

The vertical dynamics of the agents are governed by the relative error between the ground tether points when they are undeployed and a set nominal height value when deployed. These set heights can be different for each agent to minimize the likelihood of collisions for overlapping sample arrangements, similar to that in Figure 4b. However, this is unnecessary for configurations where the tether radii of the agents do not overlap at all, as in Figure 4a.

### 2.2.2. Trajectory Planning

The optimal agent waypoint is chosen using a cost function strategy that evaluates every cell of the occupancy grid map, as defined in Section 2.1. This is an appropriate method as many existing commercial-off-the-shelf (COTS) autopilots, such as Pixhawk, accept GPS coordinates as waypoints when generating UAV trajectories [20].

A receding horizon control strategy described by [16] is used. In this strategy, at every time step $t$, a subset of reachable or possible waypoints in the next $n_{RHC}$ steps for each agent is selected, as defined by their maximum velocity and turning rate (but still constrained by tether radius). Within this subset, the optimality of each cell is considered using a cost function that incorporates the target probability score first and foremost and then the difficulty for the agent to reach that position in terms of distance, heading change,

tether limits, as well as the trajectory or proximity of a target or another agent. Therefore, the optimal waypoint is defined as the cell $\bar{c}$ with a minimum cost $C$ :

$$\begin{matrix} min \\ C \end{matrix} C = \alpha{\cdot}f_{k'}(\bar{c}) + \beta{\cdot}f_d(\bar{c}) + \gamma{\cdot}f_h(\bar{c}) + \delta{\cdot}f_R(\bar{c}) + \varepsilon{\cdot}(f_{beyond}(\bar{c}) + f_T(\bar{c}) + f_{other}(\bar{c})) \quad (5)$$

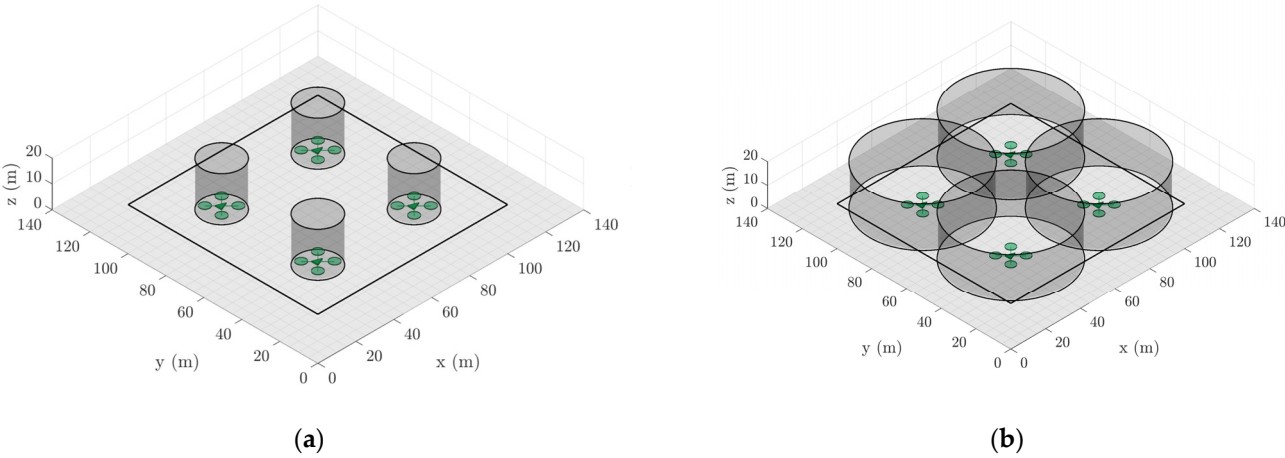

(**a**)             (**b**)

**Figure 4.** Sample arrangement involving four UAVs with different spatial constraints and operating regions imposed by their tether lengths. (**a**) No height constraints are necessary for this configuration. (**b**) The set heights should be different in this configuration to avoid collisions.

The first term in Equation (5) correlates to the complement of the target probability score, as shown in Equation (6) (the higher the probability, the lower the associated cost of that cell):

$$f_{k'}\left(\bar{c}\right) = 1 - k\left(t + n, \bar{c}\right). \quad (6)$$

The second term in Equation (5), $f_d$, corresponds to the distance cost based on a normalized horizontal distance between the centre coordinates of every reachable cell in the receding horizon and the agent's current position:

$$f_d\left(\bar{c}\right) = \frac{\sqrt{\left(\bar{c}_x - \bar{c}_{UAV_x}\right)^2 + \left(\bar{c}_y - \bar{c}_{UAV_y}\right)^2}}{n{\cdot}t{\cdot}v_{max}}. \quad (7)$$

The third term, $f_h$, corresponds to the normalized heading change cost:

$$f_h\left(\bar{c}\right) = \frac{f_\psi\left(\tan^{-1}\left(\frac{\bar{c}_y - \bar{c}_{UAV_x}}{\bar{c}_y - \bar{c}_{UAV_y}}\right), \psi_{UAV}\right)}{\pi}, \quad (8)$$

where $f_\psi$ is a function that returns the smallest angle between its two input arguments.

The next introduced term, $f_R$, correlates to a repulsion field or potential field. In this potential field, the cells that have increased in target probability score over a time step are given a higher cost, with the converse effect for cells with decreasing probability score. In other words, cells with $+\Delta k$ map to the forward direction of a moving target. This motivates agents to approach a target flock from behind or from the side in preference to head-on, which may otherwise result in a collision or affect the shepherding smoothness and cohesive flock assumptions made. This resulting repulsion–attraction field cost (mapped to [0,1]) is defined as:

$$f_R\left(\bar{c}\right) = \frac{1 + k\left(t, \bar{c}\right) - k\left(t - 1, \bar{c}\right)}{2k_{max}}, \quad (9)$$

where $k_{max}$ is the maximum probability score possible for a cell.

The last term in Equation (5) is composed of three parts and correlates to the cost associated with the cells that cannot be reached within the receding horizon $\left( f_{beyond} \right)$; the cells located outside the allowable tether radius $\left( f_T \right)$; and cells that are proximal to other agents $\left( f_{other} \right)$. The values for these functions are binary; there is a zero cost for all reachable cells but a maximum cost of 1 for unreachable cells. Note that the tether limit cost here is implemented to further discourage agent UAVs from approaching their radial limits at speed or overshooting and over-tugging the tether.

The parameters of $\alpha, \beta, \gamma, \delta$ serve as relative weights in terms of importance placed on each of the components in Equation (5), and the sum to unity is as shown:

$$\alpha + \beta + \gamma + \delta = 1. \tag{10}$$

Note that the weight parameter of $\varepsilon$ from Equation (5) is instead given a nominal value $\gg 1$ to ensure that the unreachable grid cells are never optimal after applying the rest of the cost function. The different components of the cost function can be visualized from the perspective of the left agent in the sample scenario shown in Figure 5; the cell with minimum cost after applying every term in the cost function is then designated as the optimal waypoint for that specific agent, as indicated in Figure 6. The individual weight is tuned based on the emphasis of the mission. As an example, if a smoother path is required, the heading cost weight $\gamma$ should be increased. Whereas a larger $\alpha$ will make sure the agents focus on chasing the target flocks at the expense of more unnecessary accelerations and decelerations.

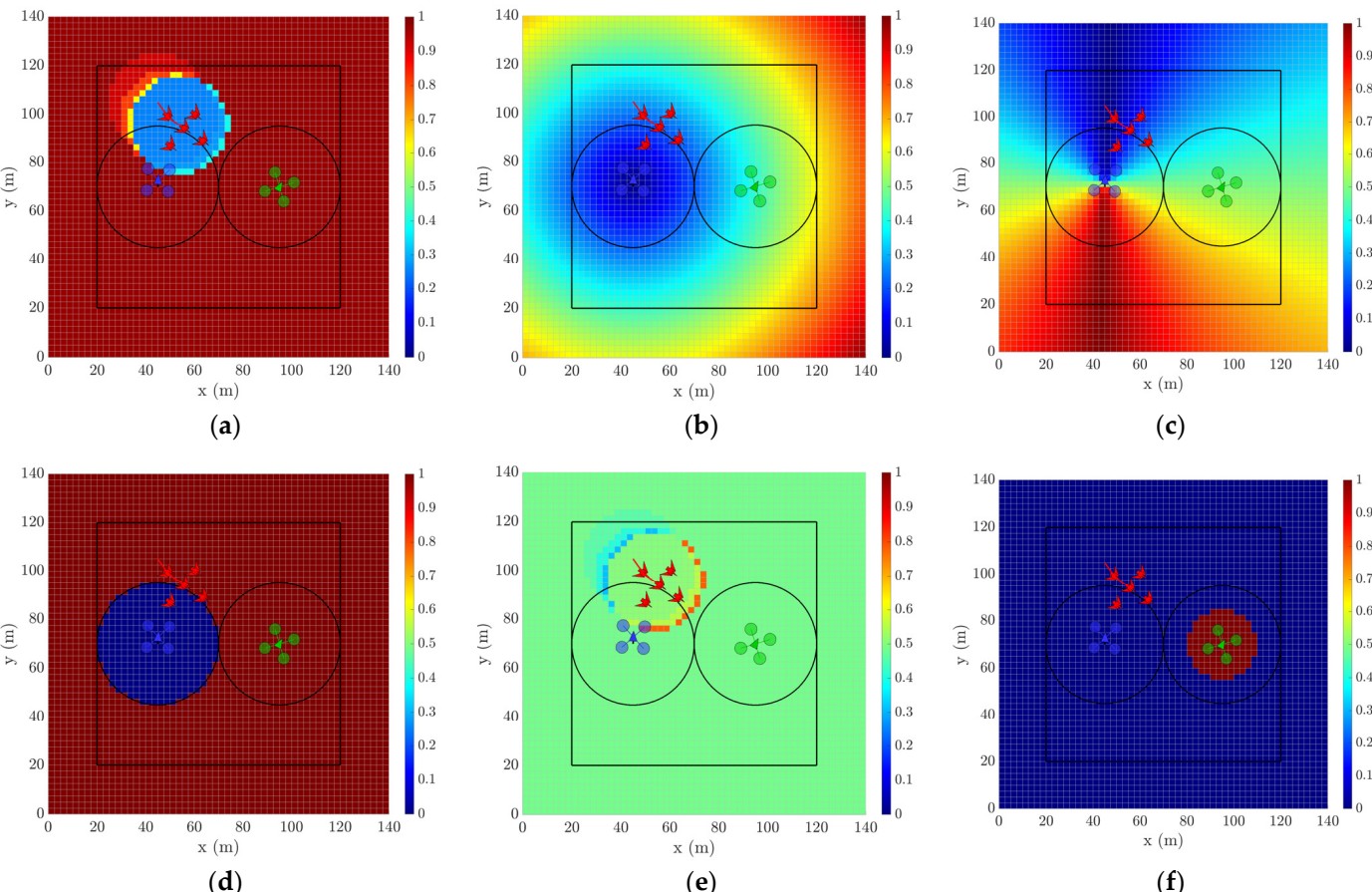

**Figure 5.** Snapshot visualization of cost function components at *t* = 5 s from perspective of the left agent. (**a**) Target probability cost; (**b**) distance cost; (**c**) heading cost; (**d**) tether limit cost; (**e**) repulsive field cost; (**f**) other agent cost.

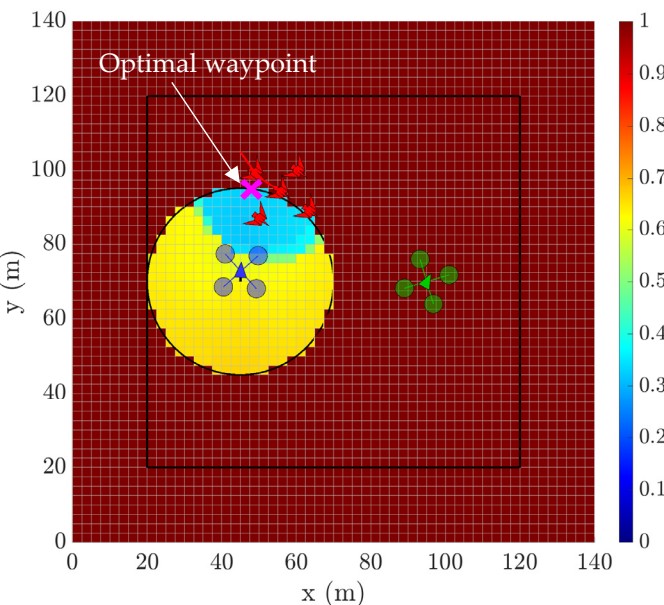

**Figure 6.** Combined cost function with weightings applied at *t* = 5 s. Optimal waypoint for left agent is shown and represents the grid cell with minimum cost.

### 2.2.3. Agent Grid Arrangement

With the dynamics and trajectory planning now defined for the agents, the next step is to determine the relative arrangement or coordinates of the agents within the property world; in other words, the ground tether points that define the centres of the pursuit region of each agent. The agent grid arrangement scheme aims to distribute *n* points as evenly as possible in a $b \times h$ sized grid, where the *n* points represent the number of agents. Centroidal Voronoi Tessellation (CVT) was chosen as a common and fairly computationally quick method to evenly partition a given plane into *n* convex polytopes [21]. CVT is a special form of Voronoi Tessellation in which the 'generating' point or node of a given partition is constrained to be coincident with its own centroid, resulting in neighbouring nodes being spaced as far apart from each other as possible.

Hence, for a set of Voronoi regions $\{V_i\}_{i=1}^{k}$, the mass centroid $c_i$ over a region with probability density $\rho(y)$ is defined as:

$$c_i = \frac{\int_{V_i} y\rho(y)dy}{\int_{V_i} \rho(y)dy}. \tag{11}$$

Lloyd's Algorithm was deemed to be appropriate for implementing and generating a CVT based on a desired number of nodes (agent number in this case) as opposed to redistributing existing nodes or data points that might be randomly distributed on a plane [21]. Note that CVTs are only defined for three or more generator points when using Delaunay triangles (the dual tessellation of the Voronoi diagram), and thus, the special degenerate cases of 1 and 2 agents were manually defined (see [21,22] for more information on CVT methods and Delaunay triangulation). The CVT method proved effective for generating a desired number of the agents' coordinates as evenly as possible within a given grid size, as shown in Figure 7.

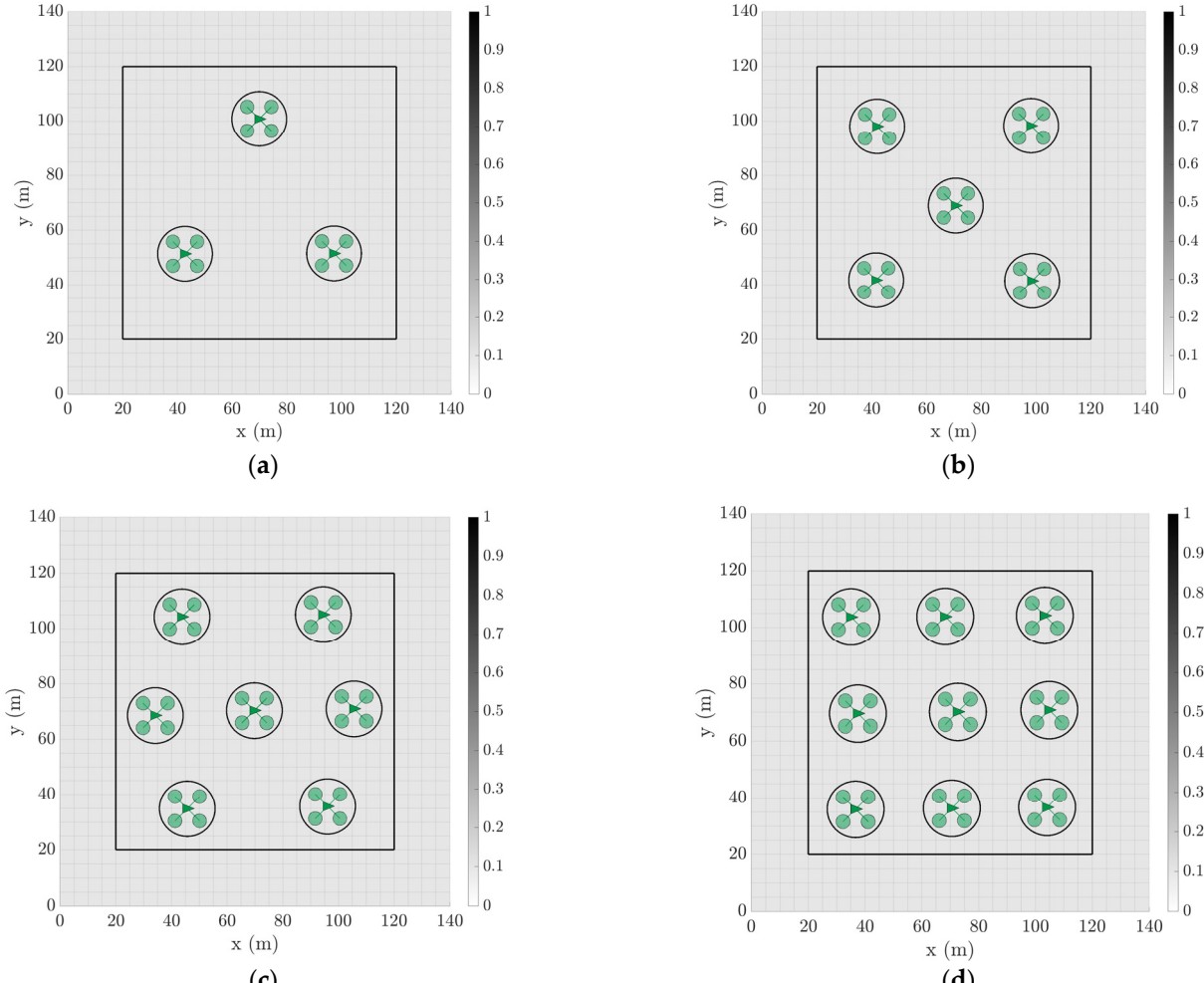

**Figure 7.** Sample grids with differing agent numbers generated using CVT method (all with $R_T = 10$ m). (**a**) $n = 3$ agents; (**b**) $n = 5$ agents; (**c**) $n = 7$ agents; (**d**) $n = 9$ agents.

### 2.2.4. Chasing Strategy and Multiple UAV Coordination

The remaining consideration for the agent UAV model is the chasing strategy and the coordination of multiple agents. A single agent can pursue its target waypoint using the cost function strategy outlined previously; however, this method becomes problematic when there are multiple agents and multiple targets since there is no hierarchy or control scheme that dictates which flock is chased by which agent.

A centralized control strategy is used to coordinate the multiple UAVs on the property. In this strategy, the coordinates of each agent at a given time step are relayed to a central computer, which then determines which agents should deploy or remain undeployed based on their proximity to high target probability areas. An agent simply needs to be within the 'scaring range', as defined by Equation (12), to be deployed:

$$r_{scare} \leq R_T + r_{no-go}, \tag{12}$$

where $R_T$ is the maximum tether length, as previously defined, while $r_{no-go}$ is the maximum radius around an agent UAV that elicits escape manoeuvres by the target flocks, as observed in experimental field trials in [7].

For the simulation, each agent can have one of the following states at any given time (colour-coded for clarity to match the simulation snapshots provided in Section 3):

- **Chase**: the agent is deployed to pursue a target flock which is in range.

- **Home**: the agent is undeployed and stationed at the ground tether point–this agent state does not scare away target flocks, irrespective of distance.
- **Return**: the agent is deployed but returning to the ground tether point since the target flock is no longer within scaring range.

In this control strategy, any in-range agents are deployed, not just the closest agent. As the position of a target is estimated using a shared probability map, an agent may pursue a midpoint position among multiple targets since it has found the geometric centre of the highest probability scores. Hence, a k-means clustering algorithm is proposed to demarcate the targets (distinct target flocks) and to help inform the agents as to which target should be pursued. The k-means algorithm is an iterative process that aims to partition a data set into k pre-defined distinct clusters through the following procedure:

1. Choose the number of clusters k.
2. Initialize centroids through random selection of k data points without replacement.
3. Compute the sum of the squared distance between all data points and every centroid.
4. Assign each data point to the closest centroid (to form distinct clusters).
5. Compute new centroids by taking the average value of data points within each respective cluster.
6. Keep iterating on steps 3–5 until converged.

In this simulation, the k value (cluster number) correlates to the number of targets and is assumed to always be known. In reality, sensor information, such as the camera detection and localisation network discussed in Section 2.1, would need to deduce how many targets are truly present. It is important to note that this procedure works for data sets with values that are spatially distributed. In a target probability map, the 'data points' or probability scores are evenly distributed since every grid cell has a value; in other words, there are no distinct clusters. Hence, to map probability score into clusters, when performing this procedure, scores below a certain threshold are removed to isolate the higher probability cells into distinct spatial clusters. Through this k-means clustering, any number of k target flocks can enter the property, and the control strategy is such that an agent will pursue the geometric centre of the closest cluster if it exists and is in range.

### 2.3. Target Flock Model

This simulation assumes that all targets are Common Starlings (Sturnus vulgaris) to reduce the number of variables and simplify the problem.

#### 2.3.1. Target Flock Dynamics and Trajectory Planning

Identical to the agent UAV model, the target flocks are modelled using simple second-order dynamics with the following state $X_{bird}$:

$$X_{bird} = (x, y, z, v_x, v_z, \psi). \tag{13}$$

Furthermore, all of the targets that make up a single target flock are assumed to remain in flock formations throughout the simulation based on field observations made in [7]. Hence, the behavioural responses and trajectory planning of the whole flock are simplified to be modelled on an individual target positioned at its geometric centre. The field observations also found that no birds were seen or remained within 50 m of a UAV. This value corresponded to the 'no-go radius' $r_{no-go}$ of Equation (12) and was grossly estimated using GPS coordinates of the UAV and observations made by the eye of the flock positions on relatively large plots of land. Hence, for this simulation with the smaller plot size of 100 m by 100 m, a conservative, reduced 'no-go' radius of 20 m has been implemented into the target flock model.

The trajectory planning of the targets is modelled using the same 'interests' system used by Wang [17]. This interests system for target flocks parallels the probability map system whereby each cell within the grid map is allocated an interest score $i$ between 0 (minimum interest) and 1 (maximum interest). The cells close to the agents are given

reduced interest scores, and a target will migrate to a cell with the highest interest once its current position falls below a certain threshold $i_{threshold}$ (to satisfy the observation that stationary birds limit movement to conserve energy). Furthermore, the cells outside of the protected area are given a very-low-interest score to ensure that the targets continuously aim to remain within the property where possible.

### 2.3.2. Target Flock Energy Expenditure System

The estimation of target flock energy expenditure in activities such as flight and foraging can serve the purpose of characterizing the perceived relative effort or energy that a target flock requires to forage in a field that is being protected by bird-deterring UAVs. An ideal bird deterrence system would aim to elicit maximum energy expenditure in a target flock to decrease the likelihood of future foraging attempts or increase the time until the given targeted flock returns to the protected property, as purported by optimal foraging theory [23,24].

Hence, a novel target flock energy expenditure system is introduced here. This model estimates an initial energy reserve dedicated for a single foraging event and makes deductions from this reserve as the target flock undergoes different states or flight modes. The important parameters that are needed to define this energy expenditure system are:

- Initial energy expenditure reserve.
- Body mass.
- Basal metabolic rate (BMR).
- Power costs for regular long flights and short burst flights.

In the literature on Common Starlings, the number of flights per day (FPD) is approximately 135 flights [25]. The average flight speed $V$ is assumed to be close to the minimum power speed as a conservative assumption; a value of approximately 10 m/s [26]. Most Starlings nests are also usually within 500 m of desirable foraging zones. Hence, the total flight time between foraging and nesting sites can be estimated by:

$$t_{flight,total} = \frac{FPD \cdot d_{travel}}{V}, \tag{14}$$

where $d_{travel}$ is the distance between a foraging zone and a nesting site. The average flight cost $P_{flight,avg}$ for the Common Starling has been estimated to be 20.4 W [27], which allows for the calculation of the allowable energy expenditure per trip for a single target flock (in J):

$$E_{exp.,single} = \frac{P_{flight,avg} \cdot t_{flight,total}}{FPD}. \tag{15}$$

The value from Equation (15) is then fed as the initial energy condition in the simulation. Note that this is a conservative case since the target flocks in the simulation initialize along the boundary of the protected property rather than at their more distant nesting sites. A summary of these parameters for the Common Starling is summarized in Table 1.

**Table 1.** Energy and mass data for the Common Starling.

| Parameter | Mass (kg) | BMR (W) | $P_{flight,avg}$ (W) | $P_{short}$ (W) | $P_{long}$ (W) |
|:---:|:---:|:---:|:---:|:---:|:---:|
| **Value** | 0.079 | 0.877 | 20.4 | 27.2 | 10.0 |

### 2.4. Simulation Testing Plan
#### Variables of Interest

The dependent variables for this simulation should indicate the performance of the bird-deterring system, and they are:

- Time taken to repel a target (if successful).
- The energy expended by a given target in a single mission. (Unsuccessful missions are capped at $t = 150$ s.)

The independent variables are determined after considering the cause and effect between different parameters:

- Number of agents.
- Tether length.
- Maximum agent speed.
- Number of targets.

At the initialization of each simulation, the target is initialized to enter at any random point along the property boundary with a heading pointing to the centre of the grid map. The simulation is then run for all the combinations of the parameters outlined in Table 2. Following this, the entire simulation process is repeated but with a different target initial position to measure the average performance of the system. Similarly, this entire simulation process is also repeated using multiple targets.

**Table 2.** Simulation parameter definitions and values.

| Variable Type | Simulation Parameters |
| --- | --- |
| Independent | Number of agents ($n = 1$–10), tether radius ($R_T = 0$–20 m), agent maximum speed ($v_{max} = 8$, 12, 16, 20 m/s), number of targets ($n_t = 1$–3) |
| Dependent | Time taken to deter target ($t$); energy expended by an individual target flock per mission ($E_{expended}$) |
| Nuisance variables and sources of randomness | Target flock entry positions, generated agent grid positions |
| Held-fixed | Property size, target flock type (Common Starling), agent controller policy, agent distribution function, agent sensor range, distribution of 'interests' (corresponding to crop type and positioning), climatic conditions, individual target experience and variation |

## 3. Simulation Results

### 3.1. Simulation Convergence Study

A simulation convergence study was performed to verify that the number of simulation repetitions is sufficient to provide a fair representation of average performance that is no longer sensitive to randomness in target entry position as well as the non-uniqueness in agent grid formations for $n \geq 3$ agents. Similarly, it is important to avoid unnecessarily large simulation repeats if results have already converged to reduce computation costs. It is assumed that the system is stable and that by the Law of Large Numbers [28], after a sufficient number of simulation repetitions, the performance of the system will converge to its true mean. Hence, a test case to observe system convergence with different configurations between $N = 1 - 500$ simulation repeats was performed (see Appendix A Figure A1). Note that the tether radius has been kept constant at a mid-range value of 14 m throughout each simulation.

This tether radius value was chosen since simulation results at the tether radius extremes were expected to converge considerably faster as the non-uniqueness issues of agent configuration and the randomness in the target entry point are diminished or less sensitive when the agents can either reach zero targets ($R_T = 0$ m) or they can reach many targets regardless of their ground tether point ($R_T = 20$ m). It was observed that there is no considerable variation in both the time performance and energy expenditure running averages after 300 simulations. Hence, it is sufficient to perform $N = 300$ simulation repeats to fairly represent the problem, and this number will be the assumed value used in the following scenarios.

### 3.2. Single Target Flock Scenarios

The scenarios in this section all involve single target flocks tested against the various agent configurations outlined in the simulation testing plan. The full averaged results are presented in Figures 8 and 9.

Observing the time performance results in Figure 8, as the tether radius $R_T$ increases, the average time taken to dispel a target flock decreases; however, this is dependent on the number of agents in the configuration as well. A successful configuration is defined as one that deters a target flock in over 90% of test cases across all the independent simulation runs, and as can be seen by the red markers, there is a 'critical' minimum tether radius for when this occurs with each agent number. For one or two agents ($n = 1 - 2$), no tether radius value allows for a successful deterrence event. Performance improves marginally for $n = 3$ agents at higher tether radius values but is still limited to a 20% success rate (at $R_T = 20$ m), while a much larger increase in performance can be seen for $n \geq 4$ agents and tether radii of $R_T \approx 13 - 14$ m—tether values above this provide diminishing improvements in performance. For this property size, the tether radius appears to have no significant impact on time performance across all values when there are $n \geq 9$ agents. The configuration that minimises the agent number and the tether radius while still remaining successful in 98% of cases at an average deterrence time of $t = 64$ s, is $n = 4$ agents and $R_T = 14$ m.

Observing the target flock energy expenditure results in Figure 9, as the tether radius $R_T$ increases, the energy expenditure in a single target bird also increases from $n = 1 - 4$ agents, after which energy expenditure decreases with increasing agent number ($n > 4$). The agent configuration that elicits maximum target energy expenditure while still satisfying the 90% success rate threshold can be seen as $n = 4$ agents with tether radii of 13 m ($E = 628$ J). Note that all the averaged energy expenditure values are still below the initialised reserve value of 1020 J.

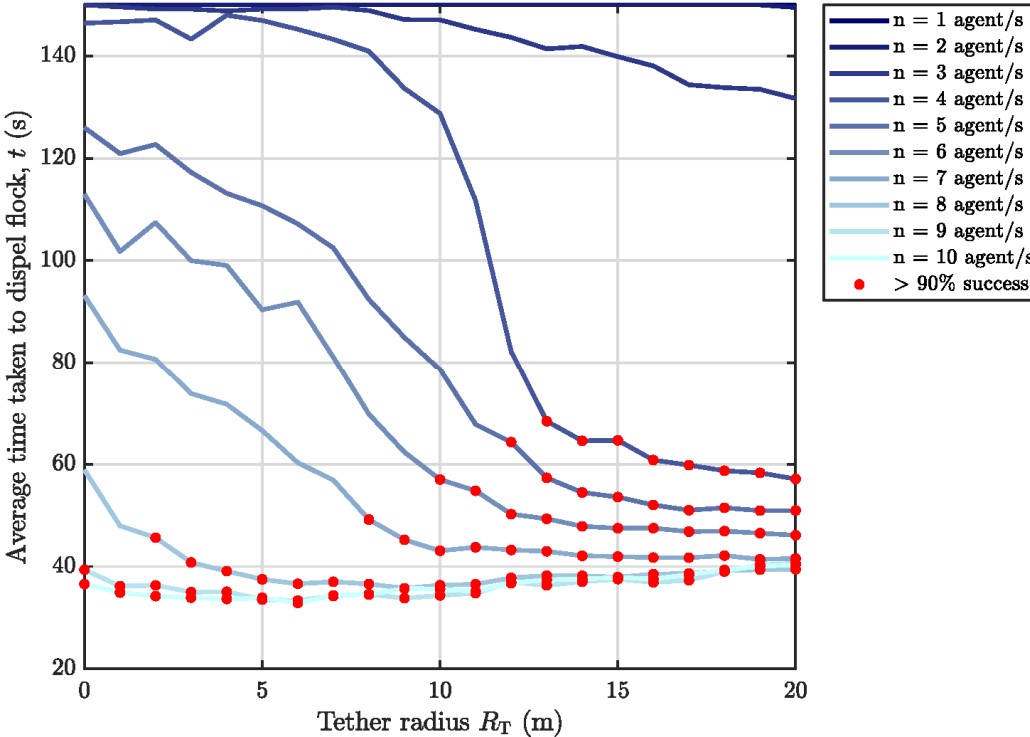

**Figure 8.** Single target flock scenarios–time performance (darker lines indicate fewer agent numbers while red markers indicate data points whose average success rate is above the given threshold across the 300 independent simulation runs). Maximum agent speed: $v_{max} = 12$ m/s.

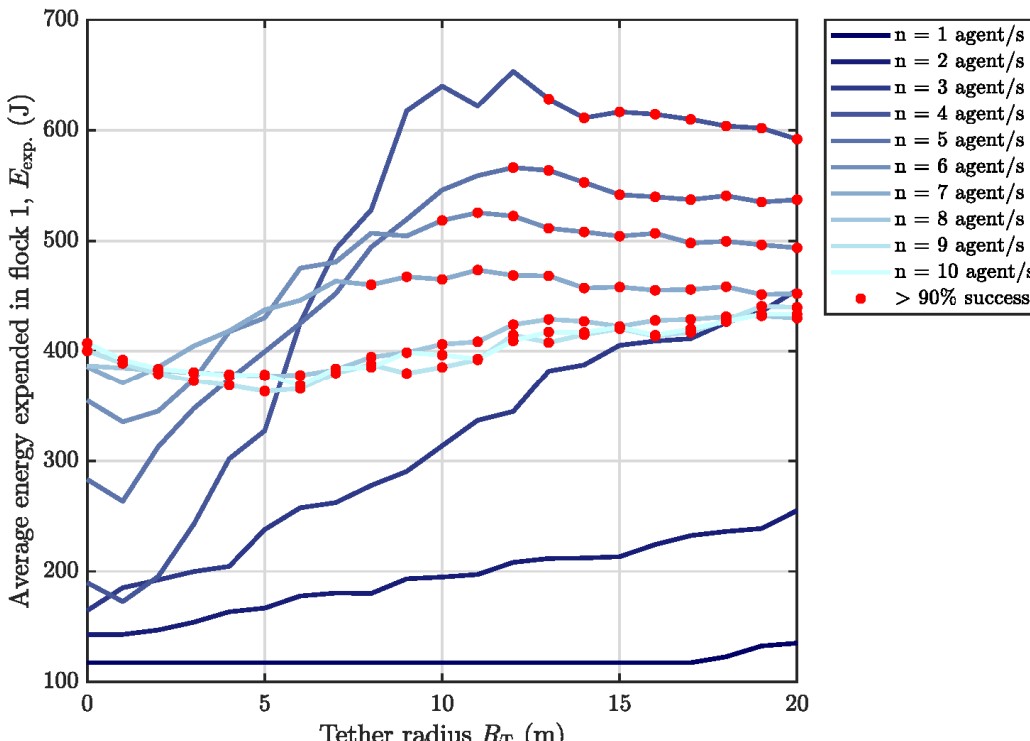

**Figure 9.** Single target flock scenarios–energy performance (darker lines indicate fewer agent numbers while red markers indicate data points whose average success rate is above the given threshold across the 300 independent simulation runs). Maximum agent speed $v_{max} = 12$ m/s.

The configuration of $n = 4$ agents and $R_T \approx 13 - 14$ m was shown to be salient; thus, snapshots for a conservative sample single target flock scenario with $n = 4$ agents and $R_T = 14$ m is shown as a reference demonstration in Figure 10 due to its marginally higher overall success rate.

In this sample simulation, the target flock is initialised with its full energy reserve at the lower left-hand side of the property (Figure 10a). After entering the property and the closest agent is deployed in its chase state (blue), the target flock seeks a waypoint on the other side of the property (Figure 10b). Once the target nears the next agent, it is deterred once again and seeks the next best interest location on the property, and this process continues to repeat (Figure 10c), with the historical agent flight paths and states shown in their respective colours. At $t = 86$ s, the target is successfully dispelled from the property and is consequently greyed out (Figure 10d). Note that its original energy reserve has been quartered by the end of the simulation. The full 3D trajectory of this simulation can be found in Appendix A Figure A2 for reference and visually captures the vertical ascents and descents that correlate to higher energy expenditure manoeuvres.

### 3.3. Multiple Target Scenarios

The scenarios in this section involve multiple target flocks tested against the same UAV configurations outlined in the single target flock scenarios above and in the testing plan. The full averaged results, run for 300 simulations on each configuration, are presented in Figures 11 and 12 ($n_T = 2$ targets). The results for $n_t = 3$ targets provided similar results, and for brevity, are omitted here and included in Appendix A Figures A3 and A4 for reference. The 2D snapshots and 3D trajectories for the triple target simulation are included in Appendix A Figures A5 and A6.

The time required to deter two target flocks is shown in Figure 11. Note that a given simulation run is not successfully complete in these scenarios until both target flocks have been dispelled. Similar trends to the single target flock scenarios can be seen here for double target flock deterrence time. The successful configuration that minimises agent number

and tether radius is evidently $n = 4$ agents and $R_T = 14$ m. This configuration provides an average deterrence time of $t = 71$ s at a success rate of 91%.

The target flock energy expenditure results are shown in Figure 12. Note that these results only constitute the values for one out of the two target flocks; conducted for the sake of brevity since the second flock results are near-identical, as expected. The best configuration eliciting maximum target flock energy expenditure ($E_{\text{exp.}} = 512$ J) is similarly $n = 4$ agents and $R_T = 14$ m, with the same success rate of 91%.

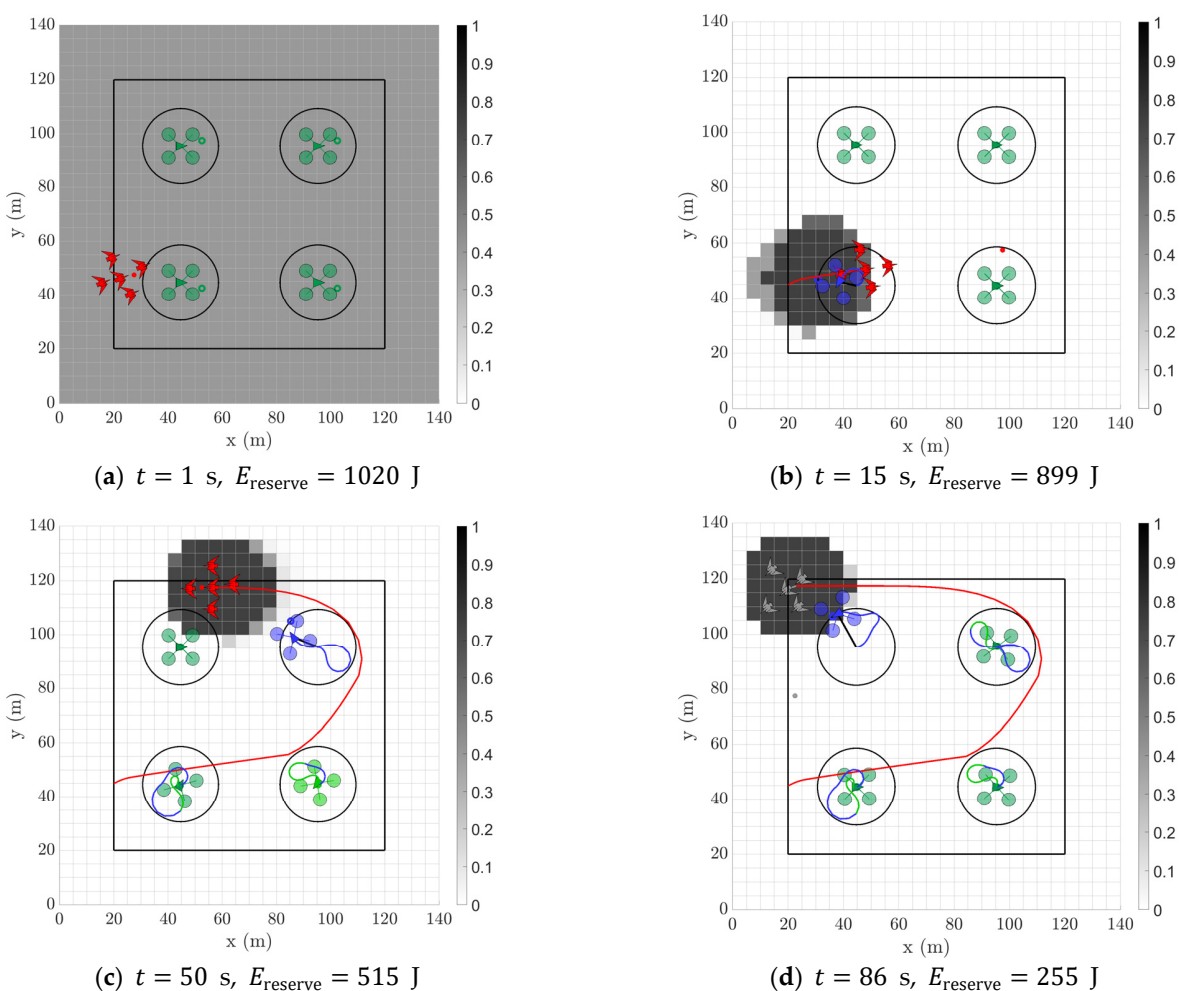

(**a**) $t = 1$ s, $E_{\text{reserve}} = 1020$ J

(**b**) $t = 15$ s, $E_{\text{reserve}} = 899$ J

(**c**) $t = 50$ s, $E_{\text{reserve}} = 515$ J

(**d**) $t = 86$ s, $E_{\text{reserve}} = 255$ J

**Figure 10.** Snapshots of a sample single target flock simulation with the following configuration: $n = 4$ agents, $R_T = 14$ m, $v_{max} = 12$ m/s. Simulation begins in (**a**) with each target in the target flock at the full energy reserve of 1020 J, all agents undeployed in their **Home** state, and each cell in the probability map initialised with the same value. As the simulation progresses from (**b**,**c**), the target flock seeks to find grid cells of maximum interest indicated by the red dot but is repelled by nearby deployed agents, shown in their **Chase** state, while agents that were deployed but no longer in range are shown in their **Return** state. Simulation is completed in (**d**) with the target centre (greyed out) no longer in the protected area by $t = 86$ s and the original energy reserve quartered.

The snapshots for a sample double target flock simulation using this configuration are presented in Figure 13. The two target flocks are initialised independently with full energy reserves along the property boundary: target 1 (red) at the bottom left-hand side and target 2 (magenta) at the bottom right-hand side (Figure 13a). After a period of time, the targets are each deterred by the closest agents in range and seek new waypoints away from the scaring stimuli (Figure 13b). Once the target flocks reach the new area in the property, they are deterred by the next pair of agents (Figure 13c). Finally, by $t = 79$ s, both targets are dispelled from the property at the same time (Figure 13d). Note that the energy reserves of

both targets are approximately halved from their initial value. The full 3D trajectory for this sample simulation is shown in Appendix A Figure A7 for reference.

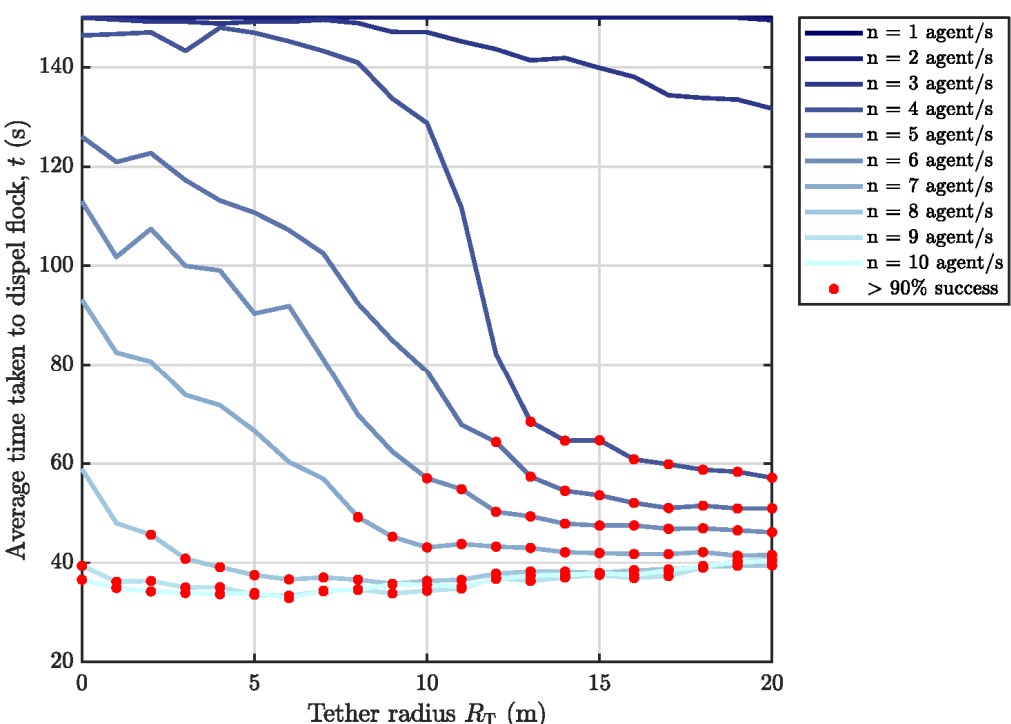

**Figure 11.** Double target flock scenarios–time performance (darker lines indicate fewer agent numbers while red markers indicate data points whose average success rate is above the given threshold across the 300 independent simulation runs). Maximum agent speed $v_{max} = 12\,\text{m/s}$.

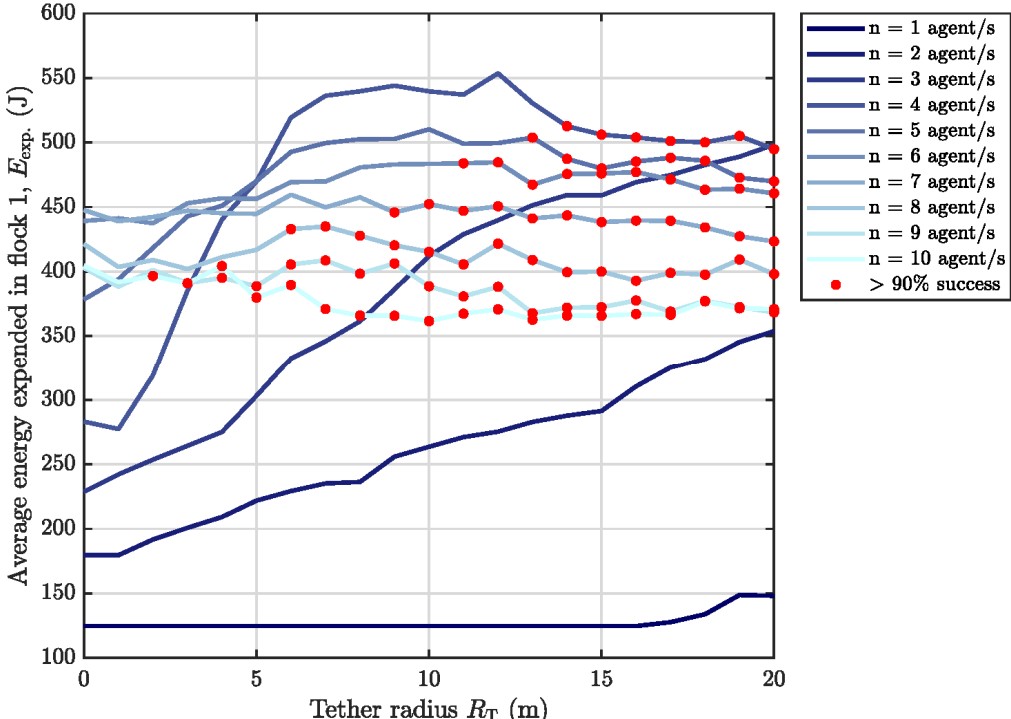

**Figure 12.** Double target flock scenarios–energy performance (darker lines indicate fewer agent numbers while red markers indicate data points whose average success rate is above the given threshold across the 300 independent simulation runs). Maximum agent speed $v_{max} = 12\,\text{m/s}$.

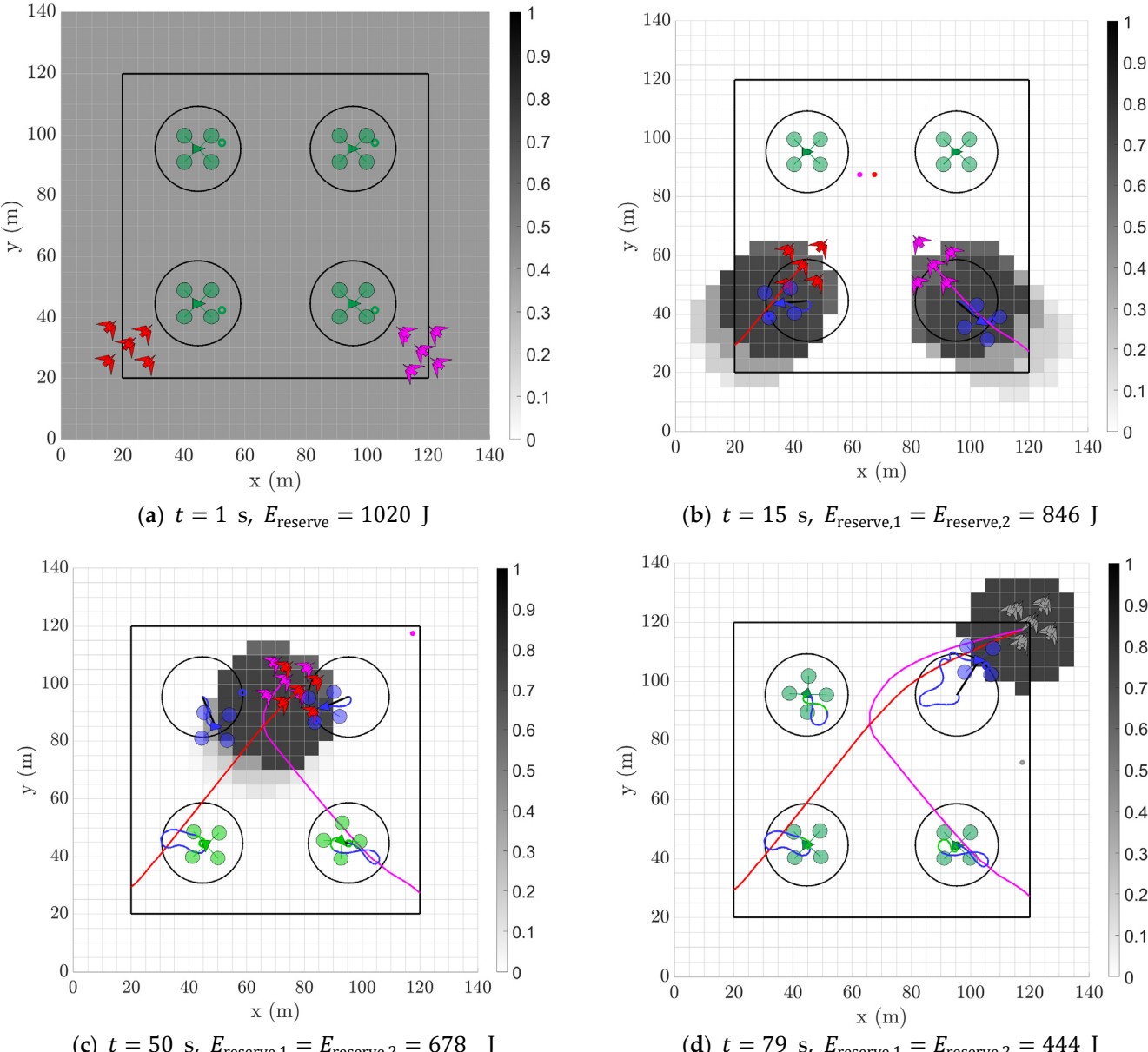

**(a)** $t = 1$ s, $E_{reserve} = 1020$ J

**(b)** $t = 15$ s, $E_{reserve,1} = E_{reserve,2} = 846$ J

**(c)** $t = 50$ s, $E_{reserve,1} = E_{reserve,2} = 678$ J

**(d)** $t = 79$ s, $E_{reserve,1} = E_{reserve,2} = 444$ J

**Figure 13.** Snapshots of a sample double target flock simulation with the following configuration: $n = 4$ agents, $R_T = 14$ m, $v_{max} = 12$ m/s. Simulation begins in (**a**) with each target in their respective target flock at the full energy reserve of 1020 J, all agents undeployed in their **Home** state, and each cell in the probability map initialised with the same value. As the simulation progresses from (**b**,**c**), both target flocks seek to find grid cells of maximum interest indicated by their respective red (**Flock 1**) and magenta (**Flock 2**) dots but are repelled by nearby deployed agents, shown in their **Chase** state, while agents that were deployed but no longer in range are shown in their **Return** state. Simulation is completed in (**d**) with both target centres (greyed out) no longer in the protected area at the same time of $t = 79$ s and the original energy reserve approximately halved.

### 3.4. Testing Agent Maximum Speed

The deterrence time and energy expenditure results for $n = 4$ agents and a single target flock at various tether radii and maximum flight speeds are shown in Figures 14 and 15, respectively. Similar performance can be seen for both metrics across all agents' maximum speeds, as was seen in previous results: as the tether radius increases, deterrence time decreases with notable improvement around the $R_T = 10 - 12$ m range, while energy expenditure increases with increasing tether radius, though this does appear to plateau

after the same $R_T = 10 - 12$ m range. Overall, there do not appear to be any salient trends or points of interest that depend on agent maximum speed, with all speed cases here tracing out near-identical performance plots.

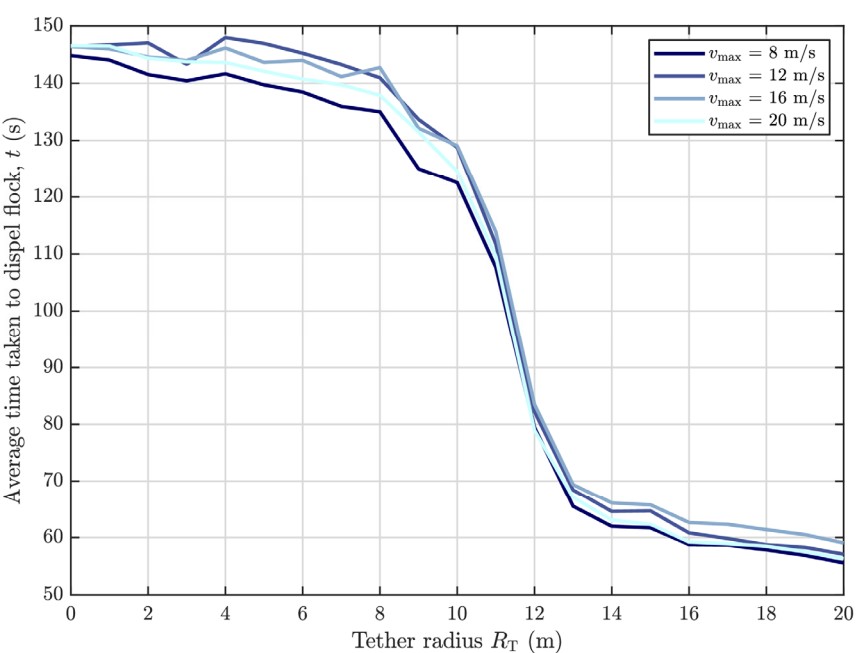

**Figure 14.** Impact of UAV maximum speed on time deterrence for a configuration of $n = 4$ agents. Darker lines indicate lower maximum speed.

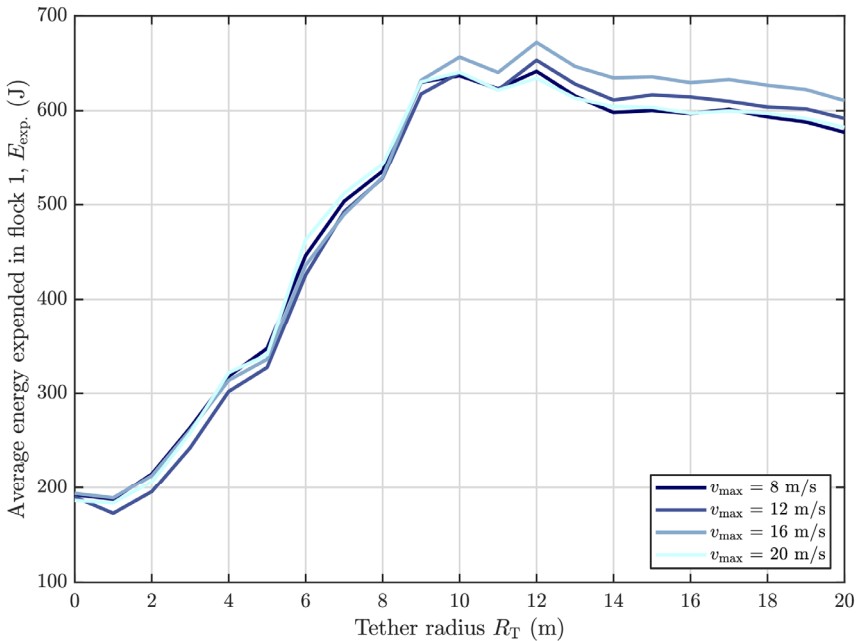

**Figure 15.** Impact of agent maximum speed on energy expenditure for a configuration of $n = 4$ agents. Darker lines indicate maximum speed.

## 4. Discussion

### 4.1. Summary and Analysis of Results

Across the single target flock and both multi-target flock scenarios, the proposed tethered UAV strategy successfully deterred the target flocks from the property area. In each scenario, the average time to deter a target flock reduced with increasing tether radius

and agent number. As was shown, every given agent number had a 'critical' tether radius value, which represented the minimum length required for at least a 90% average success rate across the full 300 simulation runs. This critical value only appears for $n \geq 4$ agents and decreases with increasing agent number, while tether radii values above the critical point provide diminishing or marginal improvements in performance. The trajectories for both the target flocks modelled via 'interests' maps and informed by field trials–as well as the pursuing agents modelled via cost functions and constrained by tether radius, are shown to be smooth. The candidate configurations that exhibited the best performance across the different target scenarios are summarised in Table 3.

**Table 3.** Simulation parameter definitions and values. Note the outlier for the case of $n_t = 2$, $R_T = 13$ m where success rate was markedly lower than other cases.

| No. Targets, $n_t$ | Tether Radius, $R_T$ (m) | Success Rate (%) |
|:---:|:---:|:---:|
| 1 | 13 | 97 |
| | 14 | 98 |
| 2 | 13 | 81 |
| | 14 | 91 |
| 3 | 13 | 91 |
| | 14 | 96 |

Considering the combination of $R_T = 13$ and $n_t = 2$, the targets only had a success rate of 81%, it is therefore ideal to opt for the 14 m version, which maintained a success rate of at least 91%; crops only need to be lightly damaged once to be downgraded in terms of quality and value; hence, the more consistently effective system should be chosen. This provides further justification for excluding the three agent configurations that only achieved a maximum success rate of 20% at $R_T = 20$ m, despite having an average deterrence time below the pseudo-infinity of $t = 150$ s in each of the time performance plots. Intriguingly, the success rate for the three target flock scenarios was *higher* than the two target flock scenarios; however, this is most likely due to the coarse modelling and the fact that having more targets means there is a higher chance to activate a nearby agent; once an agent is active, it may be more effective in chasing and funnelling the remaining targets. Note that varying the maximum agent speed for the optimal configuration, as shown in Figures 14 and 15, did not provide any salient features—all speeds provided comparable results. This is presumably because the operating regions constrained by the tether radius for each agent are quite small, and the different maximum flight speeds neither help nor hinder the overall deterrence capabilities since they are not reached in most cases.

Analysis of time performance across the different target flock number scenarios showed that the average time taken to deter all of the intruding target flocks intuitively increased with an increasing target number, seen by the general shift towards the top right in the critical minimum tether value and overall deterrence time in the plots. This indicates that longer tether radii are required to dispel more targets in an equivalent time frame when there are more targets. Though improvements in deterrence time are certainly beneficial, it is ultimately much more important that the system be functional and economically viable. Hence, though the configuration chosen provides the slowest deterrence time among those with success rates above 90%, it is preferable as the cheapest viable option.

The energy results across the different target number scenarios showed that the average target flock energy expenditure reduced and plateaued with increasing numbers of target flocks. This is presumably because when there are more target flocks, there is a higher chance that more agent UAVs are triggered for deployment, which can, in turn, have the effect of funnelling the target flocks into similar routes and positions and forcing them to take shorter flight paths across the property (compare the flight path lengths of the sample single target flock and double target flock scenarios in Figures 10 and 13, respectively). Additionally, note that the average energy expended across all scenarios and all configurations fell well below the maximum reserve value of 1020 J, indicating that all

'successful' scenarios were purely by physically repelling the targets off the property while none were through full depletion of the energy reserve, forcing the given target to stop its foraging mission in that property. Similarly, the minimum target flock energy expenditure was approximately 125 J across all scenarios, which closely matches that seen in the zero lateral movement benchmark case; this value is non-zero since the target flocks are assumed to fly in at altitude and at speed and, therefore, must consume energy to land at their first foraging location, even if it is acquired within the first few seconds upon entering.

A comparison of the optimal configuration against the free-moving agent benchmark case is shown in Table 4 and helps serve as a reference, considering that the field trial observations that informed the construction of this simulation model were on a drone. Note that a free-moving agent here is analogous to an infinite tether.

**Table 4.** Comparison between free-moving, 'infinite'-tether benchmark case ($n = 1$) with chosen tethered UAV strategy ($n = 4$). The free agent slightly outperformed in average energy expenditure elicited while the tethered team system had a better time performance metric.

| No. Targets ($n_t$) | Tether Radius $R_T$ (m) | Deterrence Time $t$ (s) | Energy Expenditure $E_{exp}$. (J) | Success Rate (%) |
|---|---|---|---|---|
| 1 | - | 65 | 677 | 100 |
|   | 14 | 65 | 611 | 98 |
| 2 | - | 103 | 662 | 94 |
|   | 14 | 71 | 512 | 91 |
| 3 | - | 113 | 599 | 89 |
|   | 14 | 78 | 491 | 96 |

For a single target flock, the tethered team configuration and free agent are on par in average deterrence time, while the free agent slightly outperforms in average energy expenditure elicited in each target and overall success rate. For the multi-target flock scenarios, the tethered system performed 31% faster than the free-moving agent in both scenarios but elicited 23% ($n_t = 2$) and 18% ($n_t = 3$) smaller energy expenditures. Importantly, however, the tethered system maintains a high success rate through to three targets.

### 4.2. Limitations of the Simulation Model

Though the tethered UAV strategy shows promise, it is important to recognise the limitations of this simulation model to ascertain the degree of confidence in how accurately it reflects reality, particularly in the following components:

- Target flock behaviour modelling limitations: as previously discussed, this factor represents the largest limitation. Within the simulation, it is assumed that targets remain within cohesive flocks the entire duration and will predictably flee from a scaring stimulus in the same manner each time and always flee to the most 'optimal' cell or spot on the property. Though some of these behaviours are modelled on true bird responses to UAVs in vineyards, it omits other behavioural factors and does not account for the fact that there can be many different bird types and species being deterred on the property at the same time, which in turn can influence decision making on where to forage if target flocks of different species do not want to be near each other. This is a fine approximation for flight initiation distance (FID) considering all bird types in the field experiments of [7] fled at distances of 50 m or greater (except for Silvereyes, which exhibit different foraging behaviours). Instead, it challenges foraging assumptions and the universality of the energy expenditure results since they are solely based on data and estimations for the Common Starling. As discussed in the energy expenditure modelling setup, this is a highly crude model due to the number of assumptions made and the scope in which each metric can vary. Ultimately, the energy model is more useful and accurate as a relative measure for identifying configurations that elicit higher expenditure costs rather than a source of absolute truth for what these expenditure values actually are. Though this simulation does

not account for returning target flocks, it is assumed by principles such as optimal foraging theory that target flocks become dissuaded the more energy they expend in searching for food on that property and will thus seek food elsewhere. This can also be beneficial and conservative for the simulation since every target flock enters with full energy reserves as opposed to depleted reserves from previous foraging attempts. A final big limitation of the target flock behaviour modelling is that it does not account for habituation; it assumes that target flocks will respond at the same distance to the scaring stimuli every time, irrespective of exposure history or perceived learning. Though this assumption may be valid for a single foraging mission, it certainly does not hold when the same target flocks are exposed to the same stimuli over periods of time.

- Agent modelling limitations: in terms of agent modelling, there were simplifications in dynamics with the hybrid 2D and 3D approach. A 3D probability map was avoided to reduce significantly higher computational costs for only marginal gains, indicating that the vertical trajectory planning for the agents was constrained to operate between two pre-set heights (the ground and 15 m), with the optimal waypoint calculated by the cost function only carrying relevance in the horizontal plane. Similarly, no dynamic considerations due to tether were modelled; it was assumed that, for the configurations under study, the horizontal offsets from the tether point were sufficiently small to not impact overall dynamics and stability to an appreciable degree, as long as there is an appropriate winch system to maintain sufficient tautness such as seen in the near-horizontal 12 m powered tether used in [13]. This adds greater pressure and motivation for a shorter tether radius, where this assumption becomes increasingly more valid. Another simplification is the assumption that agent arrangement is optimal when they are spaced as far apart from each other as possible with the motivation of maximising property area coverage. This may not be the case since most birds in field trials are observed to enter the property from one of its four sides (as opposed to directly top-down from flight), as confirmed by the higher proportion of bird crop damage normally located along the peripheral zones of the property. Hence, a more optimal configuration may account for this factor and redistribute agents closer to the property boundary accordingly. A further assumption is that the number of target flocks on the property is always known based on ground camera sensor information. The verification of this assumption lies beyond the scope of this paper. It is also important to note that the bird behavioural responses are based on a UAV that is equipped with specialised scaring stimuli (auditory and visual) and not just a regular multirotor. Hence, prototyping of the physical tethered system must incorporate and account for existing scaring stimuli.
- Property modelling limitations: in reality, most crops are not perfect squares in shape, nor are they only 100 m by 100 m in size. This simulation model further reduces the problem to an idealised scenario and assumes that all sections within the property are equally attractive to target flocks, not accounting for topographical variations nor existing pockets of potentially higher or lower foraging interest through higher local crop yields or pre-existing damage, respectively. In reality, the optimal tether radius will also need to be modified and scaled with fields of different sizes since the strategy chosen in this paper will presumably fail for much larger property dimensions.
- Probability map modelling limitations: The probability map model does not evolve in the temporal domain, nor does it vary according to environmental light and weather. At the current stage, there is not enough bird behavioural study related to the foraging activity in vineyards under different weather conditions to enable a more dynamic model. Further investigation is needed to construct a more accurate probability map model.

Nevertheless, the simulated multi-UAV tethered strategy has shown to be able to deter multiple target flocks and is a novel approach and technique not seen anywhere in the literature. The use of a team of tethered drones dedicated to bird deterrence and

simulation of target energy expenditure in response to pursuing drones are introduced here as firsts of their kind to the authors' knowledge. The simulation model has been constructed using established methods and techniques used in the literature for UAV path planning via cost function and probability map and has been benchmarked against a free-moving single agent case as performed through simulation and field trials by Wang in [7], [17]. Attempts at increasing confidence in the system have focused on the imposition of dynamic constraints based on typical multirotor capabilities as well as spatial constraints due to the tether. A configuration of four agents for a plot that is only 100 m by 100 m may seem overly costly upon first inspection, particularly if the intent is to scale up the system for larger property areas; however, it is important to remember that the scaring distances were greatly underestimated as a conservative assumption in the simulation model and that the reality may be quite different with some target flocks certainly fleeing at further distances across and within different studies. Furthermore, it is not certain that this configuration will perform as expected in a field trial, but it at least remains feasible in size and scope for testing. Hence, this returns to the original question of this paper of whether a tethered system is feasible. This simulation model has shown that it could work, even if not fully optimised, though it must be validated through physical testing.

## 5. Conclusions and Future Work

### 5.1. Future Testing

The next most important step is certainly a validation of the simulation model and the chosen configuration. This requires physical testing of sub-component features such as the multi-UAV coordination strategy as well as verification of stable flight under a tether. Ultimately, true verification, once sub-component features have been tested, is through in situ field trials with live birds. Unfortunately, restrictions imposed on the Greater Sydney region between July and October 2021 due to the COVID-19 pandemic limited all access to university laboratories as well as crops in regional areas where field trials would be performed. Hence, the scope of this research paper remains strictly theoretical via simulation modelling since all forms of physical testing were not possible. Nevertheless, various proposed tests and prototype possibilities for a future study when restrictions are eased are outlined below. Note that although features such as power-over-tether serve as driving factors for the tethered design in the first place, it is well-established that power-over-tether technologies already exist and, therefore, verification or feasibility tests for this lie outside the scope of this paper. Rather, the simulation results presented depend heavily on the success of the multi-UAV coordination strategy, the impact of the tether on dynamics being minimal, and that the target flocks respond to the tethered drone system in a similar manner to free-flying drones recorded in existing field trials. Hence, these areas must be tested first and foremost to truly evaluate whether a team of tethered drones can coordinate in a grid-confined arrangement to scare incoming target flocks away. Planned future testings include:

- Autonomous Multi-UAV Coordination Test: evaluate the coordination strategy between multiple drones using real hardware.
- Tethered Drone Dynamics Test: considering that the influence of a tether on the dynamics and stability of a drone was not explicitly modelled in the simulation, it is important to verify that its impact is near-minimal in the desired testing range of $R_T = 0 - 14$ m and up to a target altitude of 15 m.
- Bird Response to Tethered UAV Test: after the multi-UAV coordination strategy and tethered dynamics have been independently tested and analysed (with consequent changes made wherever needed), the system can be tested as a full prototype in situ on a vineyard.

*5.2. Future Research*

Target Flock Modelling

Modelling target flock behaviour predictably and accurately continues to be a challenge and will, therefore, always need to be verified through physical tests with live birds. Nevertheless, research into foraging styles and preferences could assist in nuancing which areas of a property are more at risk of bird damage. For instance, flocks of European Starlings have been shown to concentrate their feeding to habitually visited areas and completely ignore other viable spots [29], leaving some parts of a crop experiencing disproportionate amounts of damage. Similarly, the target flock energy expenditure model introduced in this paper should be further refined. Though the energy model serves more as a proxy for relative effort and target flock persistence, a catalogue of the energy and flight data for more target species seen on food crops would allow for a better comparison of different target responses to UAVs.

Furthermore, it was assumed that all target flocks were the same size and caused the same amount of damage. However, not all targets will enter and forage as part of larger flocks, and differently sized crops can influence the amount of damage or show varied reception to scaring stimuli. Hence, research should be undertaken to understand the impact of flock size on bird damage and the efficacy of UAVs in deterring them to further inform the chasing strategy of the agents. The potential research areas include:

- Multi-UAV coordination optimisation: research and development into a decentralised or distributed control architecture where each agent has the capacity to make more 'informed' decisions in chasing.
- Multi-UAV arrangement optimisation: the current simulation model assumes that every spot along the property boundary is an equally likely entry point; however, further research should explore different optimised arrangements and grid-generating algorithms which do account for local geography and areas with historically greater recorded bird damage.
- Property size and shape: importantly, further work should analyse the impact of different property sizes and shapes and determine whether there is a relationship that determines the number of drones required per square metre and the scalability of different configurations. Similarly, it should be determined whether there is a relationship between tether length and property size (for a given number of agents or property 'coverage' ratio).
- Other applications: the strategy could be analysed in other specialised or smaller contexts, such as hangars or airports where birds can similarly cause disruption, but deterrence regions may only need to be in very specific zones.

*5.3. Conclusions*

The objective of this research paper was to determine the feasibility of deterring pest birds using multiple UAVs that are individually tethered to the ground in a grid. This objective was achieved; however, the research used simulation and remains theoretical due to the COVID-19 restrictions imposed on movement and university access which prohibited physical testing and validation. The simulation model was constructed as an extension of the novel autonomous trajectory planning algorithm based on a probability map and bird field trial observations by Wang in [17]. The simulation model was then tested and benchmarked to find suitable tethered agent configurations for the bird deterrence problem that would have then been presented for physical tests. An analysis highlighted the optimal candidate configuration for the 100 m by 100 m plot as requiring four evenly distributed agent UAVs with tether radii of 14 m.

This research made the following explicit contributions to the tethered UAV bird deterrence problem:

- Development of a simulation model that can operate on a wider testing range, including number of UAVs, number of target flocks and maximum speeds.

- Introduction of tethered UAVs as a potential solution to the bird deterrence problem—implemented both as a spatial constraint and in the cost function strategy within the simulation model.
- Addition of a grid-generation algorithm to arrange any number of UAVs evenly within a given rectangular area.
- Expansion of the UAV bird deterrence problem to a pseudo-3D model involving a 2D probability map representing target flock occupancy but 3D dynamics for the UAVs and target flocks.
- Introduction of a centralised multi-UAV control strategy that can cope and coordinate the chasing of multiple incoming target flocks.
- Introduction of a novel target flock energy expenditure model based on the Common Starling used to relativise effort required by a bird in a given foraging mission due to a particular agent configuration and which can be scaled up to include a catalogue of other target species.

The tethered multi-UAV strategy for bird deterrence is a novel solution proposed in this paper for agriculture and not seen elsewhere to the authors' knowledge. Although future field experiments are essential in ultimately confirming the viability of such a system, simulations of this novel bird deterrence method still show great promise that this solution is, in fact, feasible.

**Author Contributions:** Conceptualisation, K.C.W. and Z.W.; Methodology, J.T. and Z.W.; Software: J.T. and Z.W.; Validation: J.T. and Z.W.; Formal Analysis: J.T.; Investigation: J.T.; Resources: J.T., Z.W. and K.C.W.; Data curation: J.T.; Writing—original draft preparation: J.T. and Z.W.; writing—review and editing: J.T., Z.W. and K.C.W.; visualisation: J.T.; supervision: K.C.W. and Z.W.; Project administration: K.C.W. and Z.W. All authors have read and agreed to the published version of the manuscript.

**Funding:** This research received no external funding.

**Institutional Review Board Statement:** Not applicable.

**Informed Consent Statement:** Not applicable.

**Data Availability Statement:** The source code of the simulation and the videos are open source and available at https://github.com/ZihaoUSYD/TetheredBirdDeterringDrone (Accessed on 9 March 2023).

**Acknowledgments:** We would like to thank Agent Oriented Software and CEO Andrew Lucas for the idea and support that made this project possible.

**Conflicts of Interest:** The authors declare no conflict of interest.

**Appendix A**

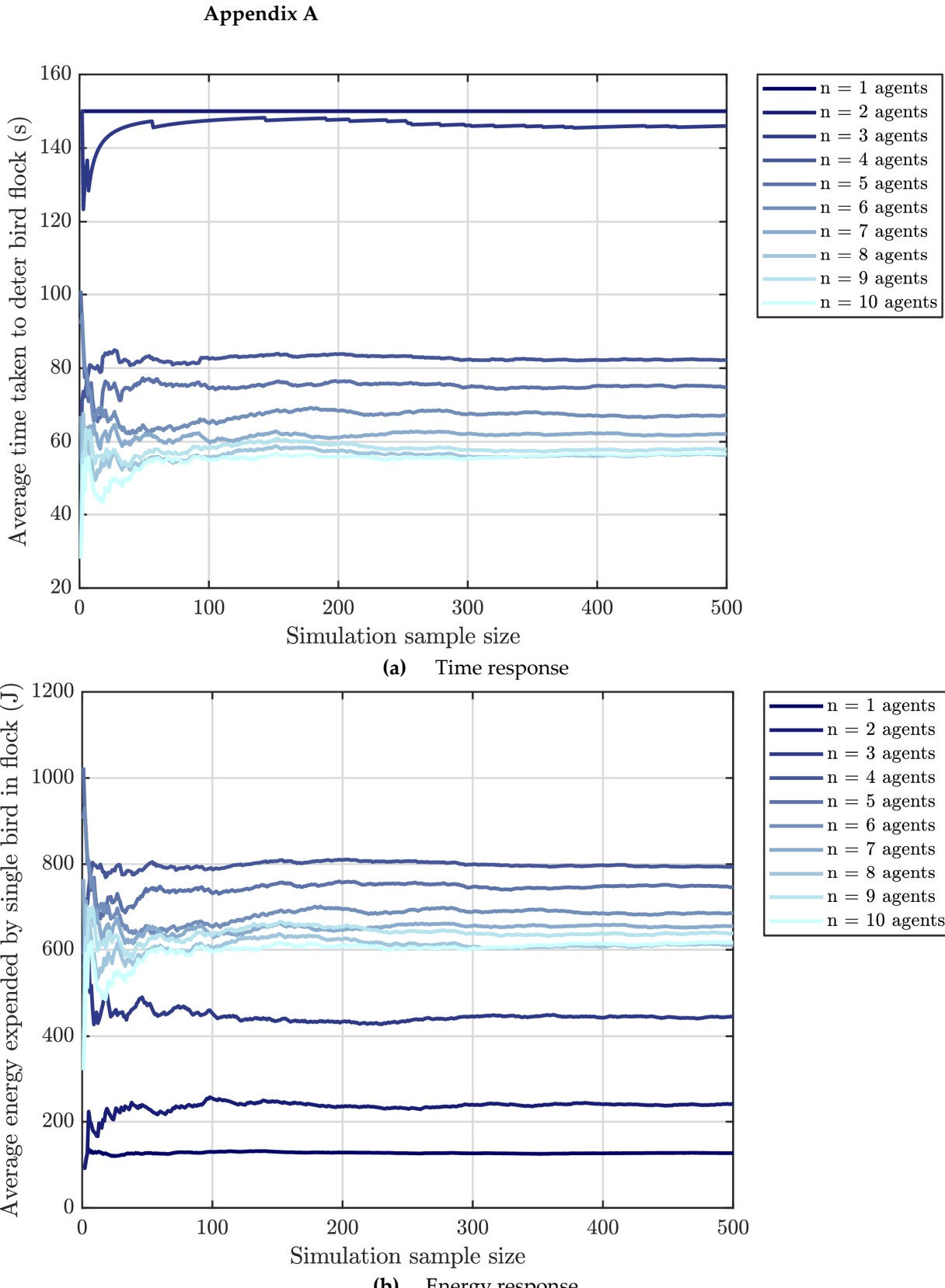

**(a)** Time response

**(b)** Energy response

**Figure A1.** Simulation convergence study for (**a**) time response and (**b**) energy expenditure, with varied agent number but fixed tether radius at $R_T = 14$ m. As simulation sample size increases, more values are being used to calculate the mean; every data point represents the running average of all preceding simulations from a minimum of 1 simulation up to a maximum of 500 simulations.

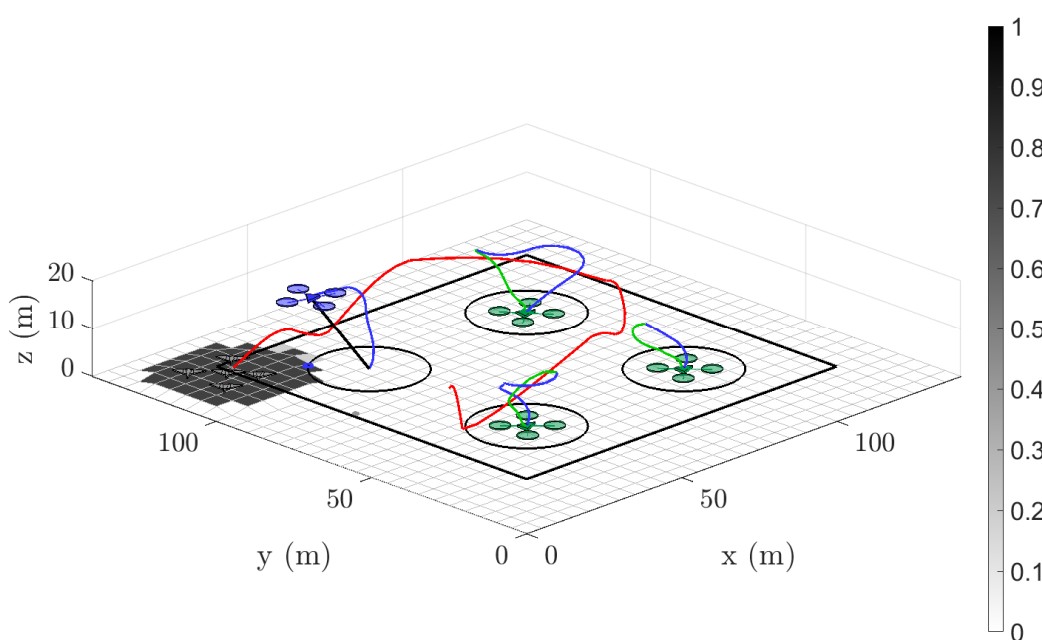

**Figure A2.** Single-target scenarios—sample simulation 3D trajectory by *t* = 86 s, $E_{reserve}$ = 255 J. Regions of maximum energy expenditure correlate to high acceleration areas such as take-offs and landings.

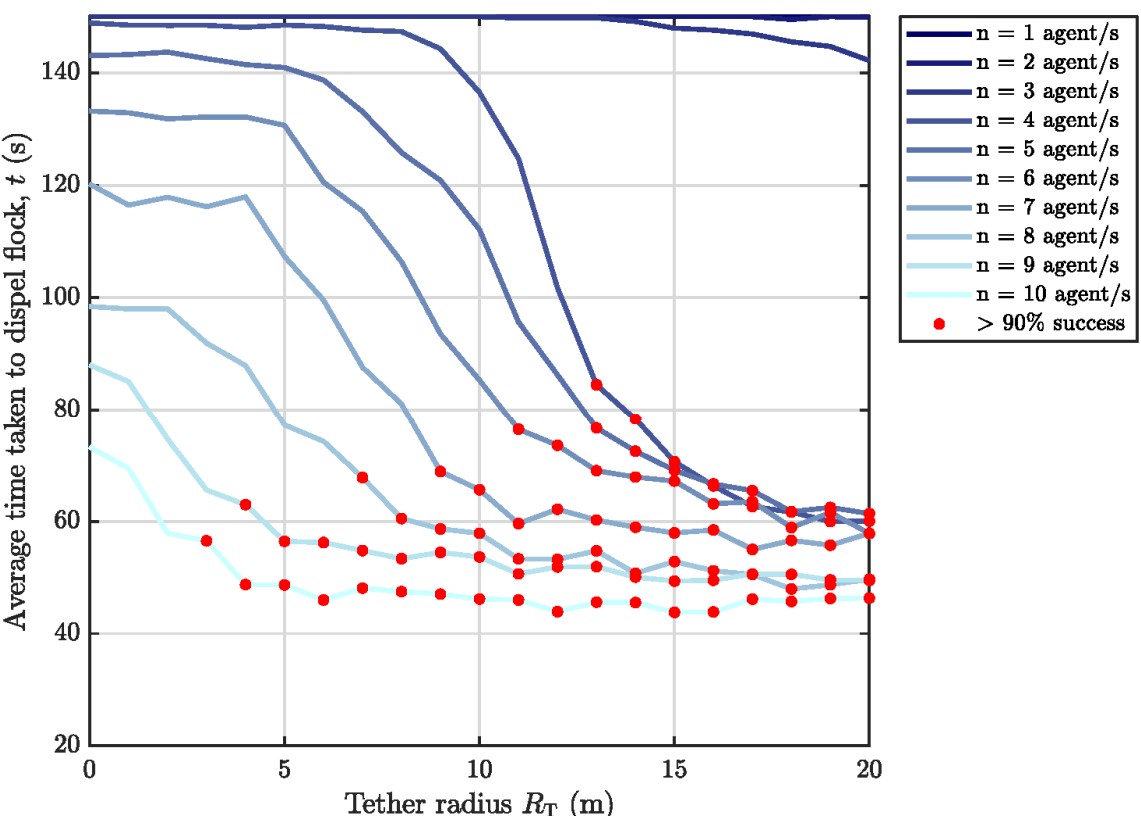

**Figure A3.** Triple-target scenarios—time performance (darker lines indicate fewer agent numbers while red markers indicate data points whose average success rate is above the given threshold). Maximum agent speed $v_{max}$ = 12 m/s.

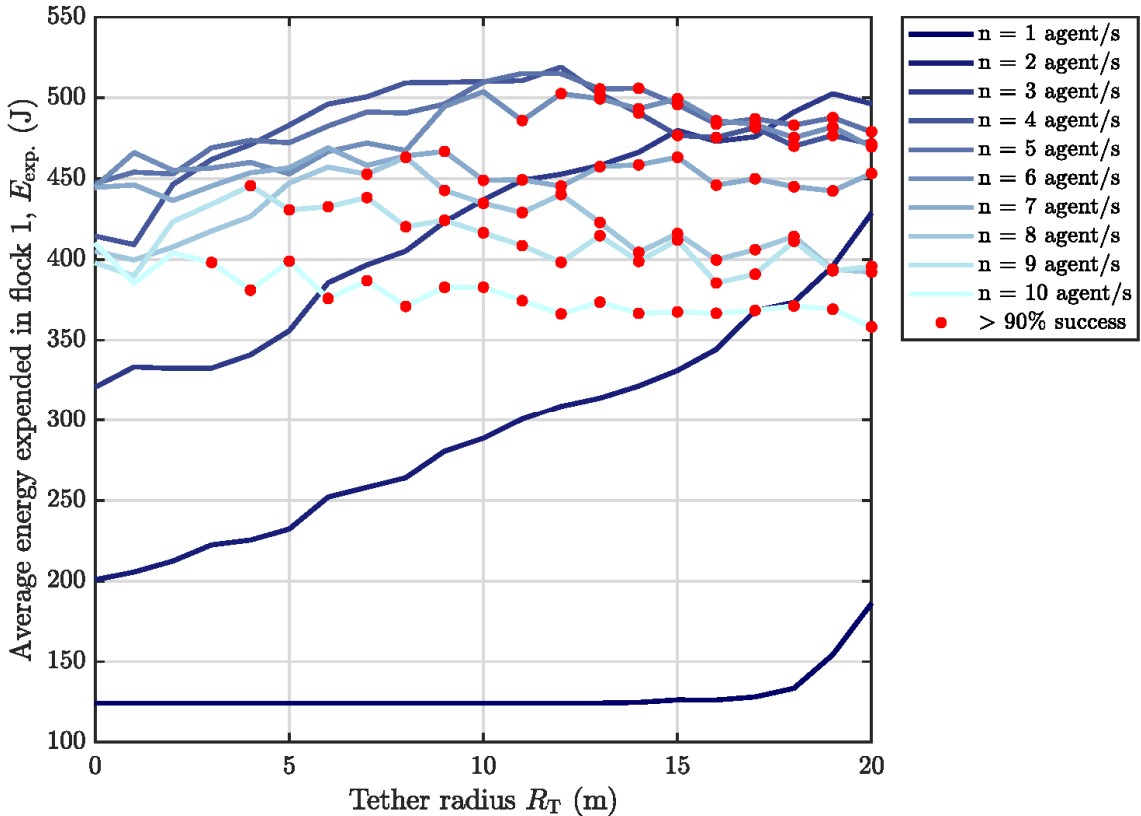

**Figure A4.** Triple-target scenarios—time performance (darker lines indicate fewer agent numbers while red markers indicate data points whose average success rate is above the given threshold across the 300 independent simulation runs). Maximum agent speed $v_{max} = 12$ m/s.

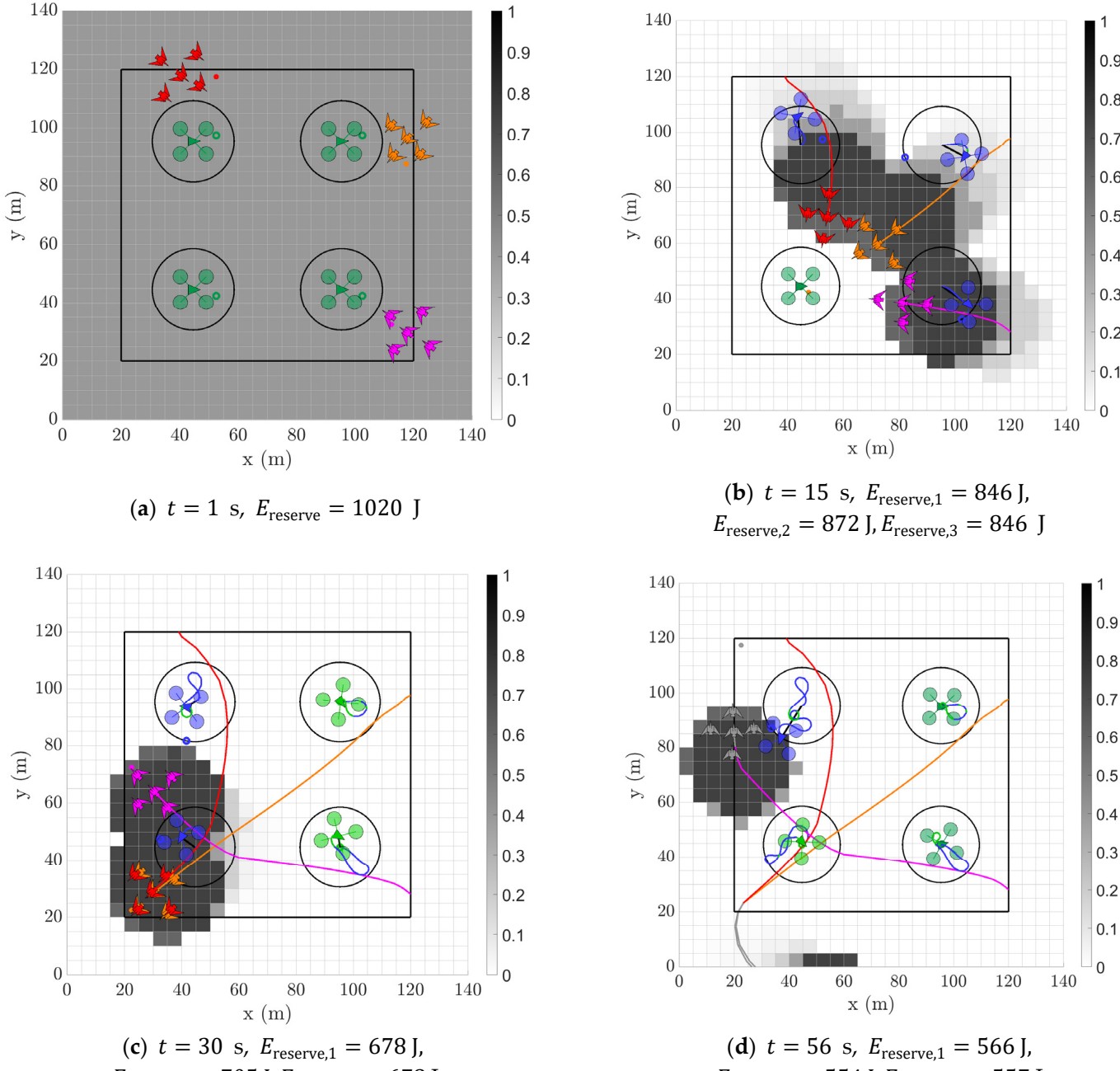

(**a**) $t = 1$ s, $E_{\text{reserve}} = 1020$ J

(**b**) $t = 15$ s, $E_{\text{reserve,1}} = 846$ J, $E_{\text{reserve,2}} = 872$ J, $E_{\text{reserve,3}} = 846$ J

(**c**) $t = 30$ s, $E_{\text{reserve,1}} = 678$ J, $E_{\text{reserve,2}} = 705$ J, $E_{\text{reserve,3}} = 678$ J

(**d**) $t = 56$ s, $E_{\text{reserve,1}} = 566$ J, $E_{\text{reserve,2}} = 554$ J, $E_{\text{reserve,3}} = 557$ J

**Figure A5.** Snapshots of a sample triple target simulation with the following configuration: $n = 4$ agents, $R_T = 14$ m, $v_{max} = 12$ m/s. Simulation begins in (**a**) with each target in the flock at the full energy reserve of 1020 J, all agentjer I u s undeployed in their **Home** state, and each cell in the probability map initialised with the same value. As the simulation progresses from (**b**,**c**), both target flocks seek to find grid cells of maximum interest indicated by their respective red (**Flock 1**), magenta (**Flock 2**) and orange (**Flock 3**) dots but are repelled by nearby deployed agents, shown in their **Chase** state, while agents that were deployed but no longer in range are shown in their **Return** state. Simulation is completed in (**d**) with both target flocks centre (greyed out) no longer in the protected area at the same time of $t = 56$ s and the original energy reserve approximately halved.

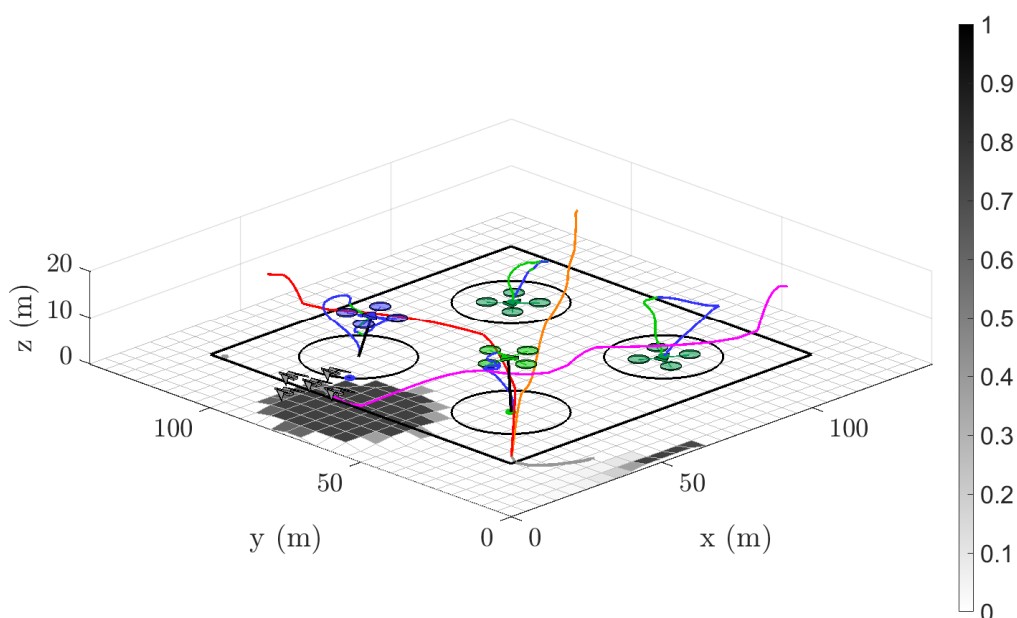

**Figure A6.** Triple-target scenarios—sample simulation 3D trajectory by $t = 56$ s, for all three target flocks. Regions of maximum energy expenditure correlate to high acceleration areas such as take-offs and landings.

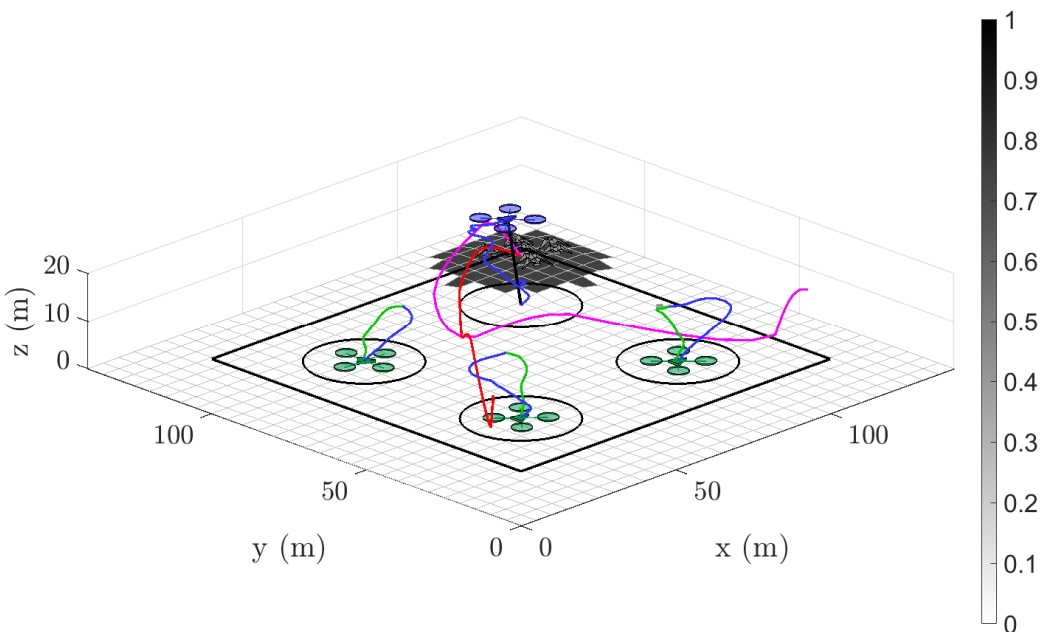

**Figure A7.** Double-target scenarios—sample simulation 3D trajectory by $t = 79$ s, $E_{\text{reserve}} = 444$ J for both target flocks. Regions of maximum energy expenditure correlate to high acceleration areas such as take-offs and landings.

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
