# Peer review of "The Viability of a Grid of Autonomous Ground-Tethered UAV Platforms in Agricultural Pest Bird Control"

_machines, doi:10.3390/machines11030377_

Round 1
Reviewer 1 Report
Summary: This paper looks at the viability of using ground-tethered UAVs to scare off pest birds in a region of interest. We found this paper to be well-motivated with a unique problem that has likely not been studied extensively. The paper does not have a lot of technical depth and reads more like an application of existing methods to a new problem. I believe the authors could do a better job of justifying the techniques they chose to use, rather than just stating the approach. The paper itself read very smoothly and overall was enjoyable and interesting to read.
We request that the authors make the following corrections/improvements to their paper:
-
The authors could be a bit more consistent in what they call the bird and UAVs. Throughout the paper, the birds are referred to as target flocks, bird flocks, flocks, birds, targets, single targets, and bird target flocks. Sometimes it wasn’t clear if birds were different from bird flocks–using the same terminology throughout would clear this confusion. Do single targets refer to a single target or a flock of birds? Also, maybe "two target-flocks" could be hyphenated to indicate that you are referring to two different flocks rather than flocks with only two birds.
-
On page 2, lines 83-87 are all those specific points supposed to be cited from ref [7]? If not, could there be a citation verifying that UAVs flying from ground level to 15 m have shown comparable success? Similarly, the statements made on lines 619-623 should be justified with citations.
-
In Equation (1) and afterward, I am not familiar with the terminology to have a vertical bar to indicate that c\bar is a vector. This should probably be explicitly stated before the equation.
-
I also dislike the use of “vertical” and “horizontal” to describe the 2D boundary axis because “vertical” brings to mind moving up/down w.r.t. the earth, rather than a flat boundary on the earth.
-
Figure 7 mistakenly repeats subgraphs a and b twice (missing the 7 and 9 agent graphs).
-
I would like to have more justification on why Equation (3) was chosen since there could be other valid models–e.g. exponential decay.
-
It is not clear how the tuning parameters in Eq (5) and Eq (3) were chosen for the simulations presented. Nor is it clear how sensitive the algorithm outcomes are to a specific choice of parameters.
-
Equations are not always followed by punctuation as they should be. Also, the sentences surrounding equations could be simplified by not stating the equation number prior to giving the equation (since that is redundant). For example for Equation (2), the sentence could read, “...of each cell is therefore defined as (followed by the equation with a period after the M\tilde equation)”. Rather than “...of each cell is therefore defined as follows in Equation 2: …“.
-
The sentence on line 294 is missing an “in” to make it grammatically correct. Overall, this paper had remarkably few grammatical/punctuation errors.
-
It wasn’t clear what the cameras on the boundary space were doing. The flock sensing is washed over as somehow being omniscient so that each UAV knows the location and number of all the flocks. This assumption seems highly unrealistic if this is expected per these boundary ground cameras. Furthermore, if the UAVs aren’t sensing the flocks, then it is unclear why a certain orientation is needed for the UAVs and why they have sensors at all. And are these cameras sensing the velocity of the flock as well?
-
I was unfamiliar with the terms “Lloyd’s Algorithm” and “Delaunay triangles” and would have appreciated a short (sentence or two) explanation of those concepts with references that point to where I could learn more.
-
The future work section is a bit too verbose. I think this could be cut down to state the main points without going into details on implementation.
-
In Table 4 it is not clear what comparison is being made. Does “free-moving benchmark” simply mean an infinite tether?
-
In Table 3, why is the success rate lower for the 2 target cases than for the three target cases?
-
In Figure 10, the Home and Return colors are hard to distinguish from each other.
-
It is not clear why the bird energy expenditure is an important metric. Why do you care how much energy they expend provided that they are scarred out of the region of interest?
-
It would be nice to have information in the abstract and introduction that discusses the overall approach. What is the “novel bird deterrence solution”? Readers have to get into the details of the later sections to piece this together. An upfront summary would be helpful.
-
It would be helpful to have an explanation of the appendices within them. They should stand a bit more on their own with analysis and a deeper explanation of the figures included in each section.
Reviewer 2 Report
The paper ‘The Viability of a Grid of Autonomous Ground-tethered UAV 2 Platforms in Agricultural Pest Bird Control’ presents a novel bird deterrent solution in the form of tethered UAVs which are attached and arranged in a grid-like fashion across a vineyard property. The simulation model successfully isolated configurations that are able to deter single and multiple incoming bird flocks using a centralized multi-UAV control strategy.
The manuscript is interesting, clear, and well-written; the following points should be clarified:
- In the real case, how much do you think could be the interference of the environmental light on the probability score value k? Please specify in the tet.
- The tether radius Rt m has been chosen in the field to be protected (100 mx100 m); do you think that, in the real case, the value of the tether radius should be modified according to the field dimensions? Please specify it.
- Pag. 9 you write that the k value of clusters is assumed to be known and correlates the number of target bird flocks. Please specify how could you get this value in real cases.
- The Introduction should be extended to give a more detailed state of the art.
